# A genome-wide nucleosome-resolution map of promoter-centered interactions in human cells corroborates the enhancer-promoter looping model

Arkadiy K Golov[1,2], Alexey A Gavrilov[1], Noam Kaplan[2]*, Sergey V Razin[1,3]*

[1]Institute of Gene Biology, Russian Academy of Sciences, Moscow, Russian Federation; [2]Department of Physiology, Biophysics & Systems Biology, Rappaport Faculty of Medicine, Technion - Israel Institute of Technology, Haifa, Israel; [3]Faculty of Biology, Lomonosov Moscow State University, Moscow, Russian Federation

## eLife Assessment

Identifying chromatin interactions with high sensitivity and resolution at the genome-wide scale continues to be technically challenging. This study introduces findings based on the improved MNase-based proximity ligation method, MChIP-C, which enables genome-wide measurement of chromatin interactions at single-nucleosome resolution. The evidence presented in this manuscript is **convincing**, and the technological advancements will be **valuable** for the study of 3D genome architecture.

**\*For correspondence:**
noam.kaplan@technion.ac.il (NK);
sergey.v.razin@gmail.com (SVR)

**Competing interest:** The authors declare that no competing interests exist.

**Abstract** The enhancer-promoter looping model, in which enhancers activate their target genes via physical contact, has long dominated the field of gene regulation. However, the ubiquity of this model has been questioned due to evidence of alternative mechanisms and the lack of its systematic validation, primarily owing to the absence of suitable experimental techniques. In this study, we present a new MNase-based proximity ligation method called MChIP-C, allowing for the measurement of protein-mediated chromatin interactions at single-nucleosome resolution on a genome-wide scale. By applying MChIP-C to study H3K4me3 promoter-centered interactions in K562 cells, we found that it had greatly improved resolution and sensitivity compared to restriction endonuclease-based C-methods. This allowed us to identify EP300 histone acetyltransferase and the SWI/SNF remodeling complex as potential candidates for establishing and/or maintaining enhancer-promoter interactions. Finally, leveraging data from published CRISPRi screens, we found that most functionally verified enhancers do physically interact with their cognate promoters, supporting the enhancer-promoter looping model.

## Introduction

For decades since the discovery of the first enhancers, the enhancer-promoter (E-P) looping model, in which enhancers activate their target genes via physical contact, has dominated the field (*Popay and Dixon, 2022*; *Ptashne, 1986*). However, direct empirical evidence for this hypothesis has been scarce, and only a handful of credible enhancers were demonstrated to be in physical proximity to their promoter partners. These include classical enhancers and their established targets such as globin genes, HoxD cluster genes, Shh and PAX6 genes (*Tolhuis et al., 2002*; *Davies et al., 2016*; *Oudelaar et al., 2019*; *Hua et al., 2021*; *Montavon et al., 2011*; *Symmons et al., 2016*; *Freire-Pritchett et al.,*

*2017*). Simultaneously, some recent microscopy observations do not seem to align well with the E-P looping model (*Benabdallah et al., 2019*; *Karr et al., 2022*).

The development of genomic assays based on proximity-ligation of chromatin, commonly known as C-methods (*Cullen et al., 1993*; *Dekker et al., 2002*; *Denker and de Laat, 2016*), has revolutionized the study of genome spatial organization, providing a complementary view to microscopic approaches (*Denker and de Laat, 2016*; *McCord et al., 2020*; *Jerkovic and Cavalli, 2021*). Both the resolution and throughput of genome-wide C-methods were instrumental in defining salient structural features of mammalian chromatin, including genomic compartments, topologically associating domains (TADs) and cohesin-dependent CTCF point interactions (*Lieberman-Aiden et al., 2009*; *Nora et al., 2012*; *Sexton et al., 2012*; *Dixon et al., 2012*; *Rao et al., 2014*; *Rowley and Corces, 2018*). However, unlike CTCF-based interactions, very few specific E-P interactions were observed with conventional restriction enzyme-based C-methods. Initially, sequencing depth inherently limited the resolution of all vs all approaches such as Hi-C, and thus a number of targeted approaches were developed. In-solution hybridization (CaptureC, CaptureHi-C) or immunoprecipitation (ChIA-PET, HiChIP, PLAC-seq) are exploited in these methods to enrich for interactions of specific regions of interest, thus decreasing the sequencing burden and allowing higher resolution (*Hughes et al., 2014*; *Mifsud et al., 2015*; *Fullwood et al., 2009*; *Mumbach et al., 2016*; *Fang et al., 2016*). While targeted methods clearly revealed a statistical preference of promoters to interact with enhancer-like regions and enabled detection of some individual E-P interactions, in some cases functionally verified enhancers did not show spatial proximity to the promoters of their targets as measured with C-methods (*Gupta et al., 2017*; *Benabdallah et al., 2019*), raising further questions concerning the universality and functional relevance of E-P looping.

Although targeted methods have circumvented sequencing-depth limitations, the resolution of C-methods based on restriction endonuclease digestion cannot overcome the inherent restriction fragment length limitation, which often results in a high level of background noise and the inability to distinguish CTCF-based interactions from other types of chromatin spatial interactions (*Goel and Hansen, 2021*). The introduction of MNase-based C-methods, such as Micro-C and its targeted oligonucleotide hybridization-based counterparts, MCC, Tiled-MCC, and RCMC, reduced the resolution limitation, allowing finer chromatin structures to be distinguished (*Hsieh et al., 2020*; *Krietenstein et al., 2020*; *Hua et al., 2021*; *Aljahani et al., 2022*; *Goel et al., 2022*). Crucially, the usage of MNase also allowed milder detergent pretreatment of crosslinked cells, increasing the sensitivity of Micro-C, MCC and related techniques by retaining E-P interactions, which appear to be more susceptible to experimental conditions than CTCF-based chromatin interactions (*Canver et al., 2015*; *Fulco et al., 2016*; *Gasperini et al., 2020*). In parallel, the development of genomic and epigenomic CRISPR-based functional screens has greatly expanded the amount of available data on functional E-P interactions. Taken together, these advancements now open the door for the systematic investigation of the spatiotemporal mechanisms underlying distal control of mammalian transcription.

Here, we present a new MNase-based C-method, MChIP-C, which allows measuring promoter-centered interactions at single nucleosome resolution on a genome-wide scale. We applied MChIP-C to construct a nucleosome-resolution map of promoter-centered interactions in human K562 cells. Taking advantage of the significantly improved resolution and sensitivity of our approach compared to restriction endonuclease-based C-methods, we investigated the molecular underpinnings of promoter-centered chromatin spatial interactions and identified EP300 histone acetyltransferase and the SWI/SNF remodeling complex as candidates for establishing and/or maintaining E-P spatial contacts. Finally, by comparing our chromatin interaction data with previously published results of CRISPRi screens (*Fulco et al., 2019*; *Gasperini et al., 2019*), we found that most functionally verified enhancers do interact with their cognate promoters, supporting the E-P looping model of enhancer activity and suggesting that in general physical interactions are central to enhancer activity in mammalian cells.

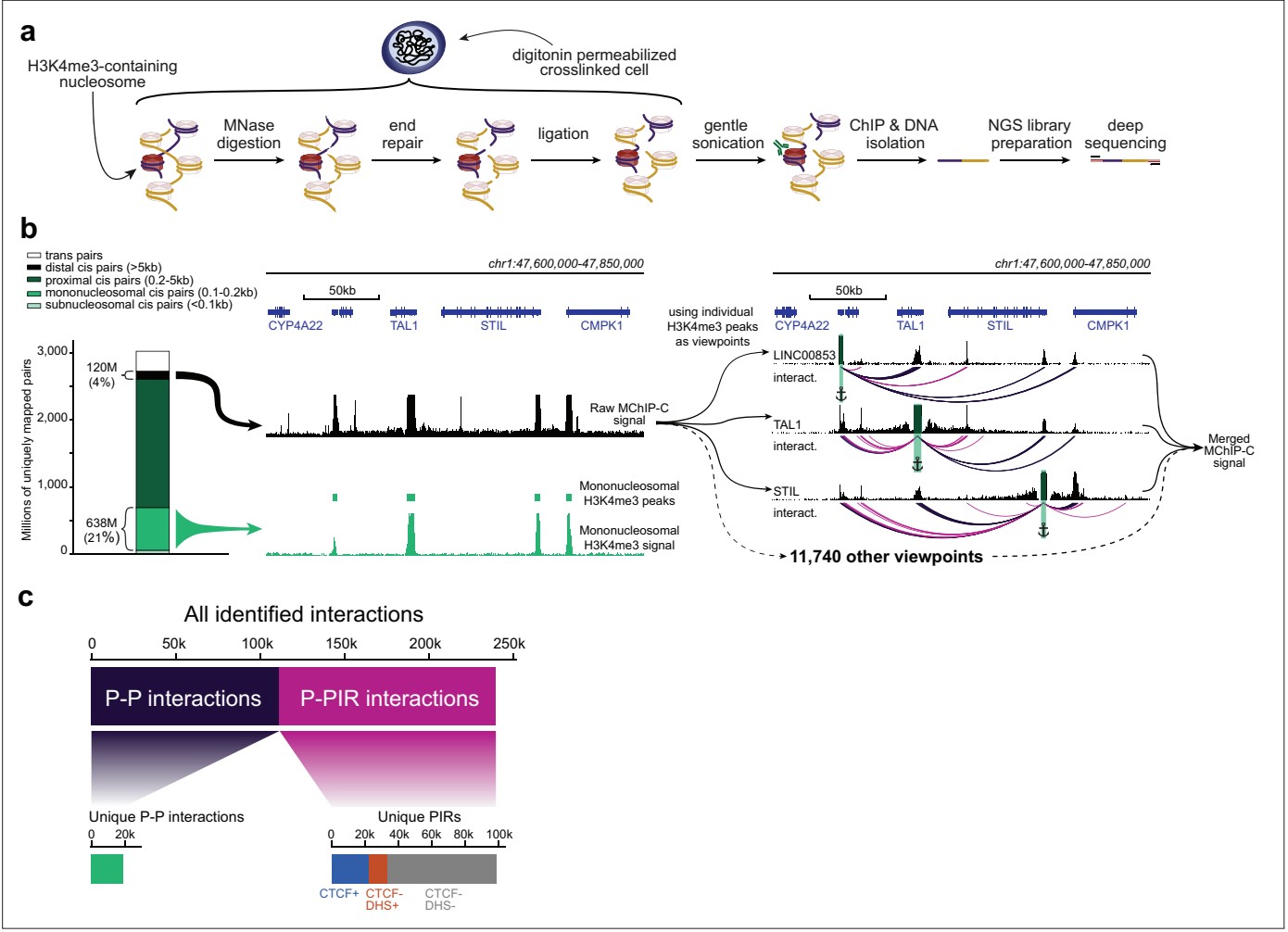

**Figure 1.** MChIP-C experimental and computational workflow. (**a**) An overview of the MChIP-C experimental procedure. (**b**) General MChIP-C analysis pipeline. A 250 kb genomic region surrounding the TAL1 gene is shown with H3K4me3 MChIP-C profiles in K562 cells. Positions of individual viewpoints are highlighted by green rectangles and anchors. Identified MChIP-C interactions are shown as magenta (P-PIR) and dark violet (P-P) arcs. (**c**) Summary statistics for all 241,073 promoter-centered MChIP-C interactions identified in K562 cells.

The online version of this article includes the following source data and figure supplement(s) for figure 1:

**Figure supplement 1.** H3K4me3 MChIP-C experiment technical assessment.

**Figure supplement 1—source data 1.** Uncropped image for gel shown in *Figure 1—figure supplement 1a* (original and with bands labelled).

## Results

### MChIP-C: antibody-based targeted measurement of genome architecture at single-nucleosome resolution

We developed MChIP-C, a novel MNase-based proximity ligation technique which combines the genome-wide throughput of HiChIP with the exceptional resolution and sensitivity of MCC (*Figure 1a*). The protocol starts similar to MCC, where cells are crosslinked with formaldehyde and permeabilized with digitonin. This milder cell permeabilization procedure has been shown to provide higher sensitivity for detecting E-P interactions (*Hua et al., 2021*). Next, chromatin is digested with micrococcal nuclease, DNA ends are blunted and proximity ligated. This is followed by sonication and chromatin immunoprecipitation with a specific antibody, as in HiChIP or PLAC-seq (*Figure 1a*). Crosslinks are then reversed, and DNA is purified, followed by library preparation and sequencing. As in MCC, we avoided using biotin to enrich for ligated fragments, potentially increasing library complexity, but also resulting in a low proportion of informative chimeric read pairs in the sequencing libraries. To partially

mitigate this, we decreased the level of DNA fragmentation and selected for longer DNA fragments (see Methods).

We performed four biological replicates of MChIP-C experiment on K562 cells using anti-H3K4me3 antibodies to focus on spatial interactions of active promoters. The MChIP-C libraries were paired-end sequenced with a total of ~3.5 B read pairs, yielding ~3 B uniquely mapped reads (*Figure 1b*; *Figure 1—figure supplement 1b*; *Supplementary file 1*). ~91.5% (~2.77 B) of these mapped within 1 kb of each other and ~4% (~120 M) mapped further than 5 kb away from each other. While the 4% fraction of informative reads was low relative to restriction enzyme-based C-methods (e.g. >60% informative reads in PLAC-seq), it is comparable to MCC (0.5–5% informative reads, see Methods).

After read mapping, we identified active promoters by first selecting nucleosome-size convergently mapping read pairs to obtain H3K4me3 occupancy profiles. We found that the profiles of replicates were highly similar (Pearson correlation 0.95–0.97) and match standard H3K4me3 ChIP-seq (Pearson correlation 0.90–0.92; *Figure 1—figure supplement 1c and d*). Next, we identified 11,743 consensus peaks amongst replicates and found these mostly colocalized with active RNA polymerase II TSSs (*Figure 1—figure supplement 1e*; *Supplementary file 1*).

We then proceeded to map promoter interactions. We defined the H3K4me3 peaks as interaction viewpoints and selected the reads mapping to each of these viewpoints, resulting in 11,743 4C-like viewpoint interaction profiles (interactively browsable online, see Data Availability). To gain a general overview of the data, we summed the profiles of all viewpoints into a single genome-wide profile which we refer to as 'merged MChIP-C' (*Figure 1b*). Both viewpoint-specific and merged MChIP-C correlated well between replicates (Pearson correlation 0.83–0.92 for merged profiles and 0.62–0.75 for viewpoint-separated profiles; *Figure 1—figure supplement 1f*), and therefore we combined all replicates, resulting in single-nucleosome resolution interaction profiles for 10,955 active promoters, after filtering low-coverage viewpoints.

The obtained interaction profiles demonstrate that promoters interact with each other and with non-promoter regions (*Figure 1b*). We systematically identified localized interactions by searching for 250 bp bins with significantly higher than expected MChIP-C signal. For each of the replicates, we found that 70–87% of its interactions were present in at least one other replicate (corresponding to the amount of sequencing per replicate; see Methods), suggesting that called interactions were generally reproducible. Overall, we detected 112,087 promoter-promoter bin (P-P) interactions and 128,986 promoter-nonpromoter bin interactions (*Figure 1c*; *Supplementary file 1*). Both types of interactions were mostly localized within TADs (*Figure 1—figure supplement 1h*). We excluded P-P interactions from subsequent analyses and focused on 128,986 promoter-nonpromoter interactions linking 10,721 viewpoints with 99,315 unique nonpromoter bins (*Supplementary file 1*) which we refer to as PIRs (Promoter Interacting Regions).

## MChIP-C provides a sensitive genome-wide view of promoter-centered interactions

We then asked how MChIP-C compares to similar approaches based on standard restriction enzymes. HiChIP and PLAC-seq are directly comparable to MChIP-C, as both HiChIP and PLAC-seq combine Hi-C with chromatin immunoprecipitation. Specifically, we compared our data to those of *Chen et al., 2022*, who used PLAC-seq with anti-H3K4me3 antibodies in K562 cells. We first visually compared the PLAC-seq and MChIP-C proximity ligation profiles of a number of genes with well-described regulatory landscape in K562 cells: MYC (*Fulco et al., 2016*; *Lin et al., 2022*), GATA1 (*Fulco et al., 2016*; *Fulco et al., 2019*), HBG2 (Moon and Ley 1991; *Liu et al., 2017*), MYB (Xie et al. 2019), and VEGFA (Aran et al. 2016; Dahan et al. 2021). We found that MChIP-C profiles showed clear highly-localized interaction peaks corresponding to the CTCF-bound sites and enhancers known to control the expression of these genes (*Figure 2a*; *Figure 2—figure supplement 1a*). For instance, the MYC gene has seven well-characterized K562-specific enhancer elements (e1-e7) spread along the 2 Mb region downstream of the gene and MChIPC allowed detection of interactions between MYC promoter and five of these sites. PLAC-seq profiles are by contrast much noisier and lack clear peaks corresponding to either CTCF sites or known enhancers of the interrogated genes. In addition, we visually compared standard Micro-C data previously reported for K562 (*Barshad et al., 2023*) to our MChIP-C data at these five loci, using only the promoter-centered interactions from the Micro-C dataset (see Methods). Interestingly, while the

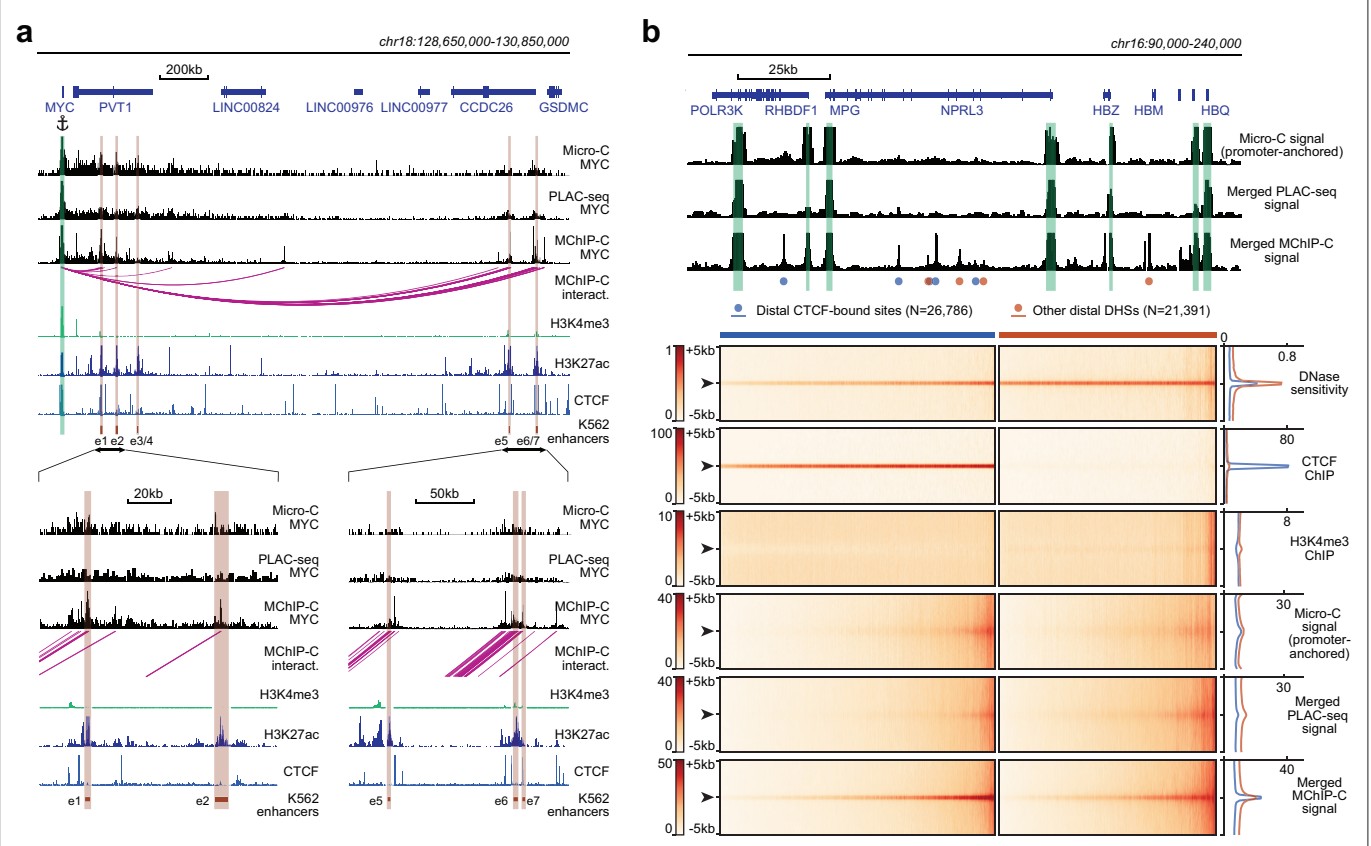

**Figure 2.** Comparison of MChIP-C with PLAC-seq and Micro-C. (**a**) Top: MChIP-C, PLAC-seq and Micro-C interaction profiles of the MYC promoter in K562 cells. MChIP-C interactions of the MYC promoter are shown as magenta arcs. Positions of 7 (e1–e7) CRISPRi-verified K562 MYC enhancers are highlighted as orange rectangles. Bottom: zoom in on two enhancer clusters. (**b**) Systematic comparison of merged MChIP-C, merged PLAC-seq and merged promoter-anchored Micro-C signals in distal regulatory sites. Top: Merged MChIP-C, merged PLAC-seq and merged promoter-anchored Micro-C profiles in a 150 kb genomic region surrounding the α-globin gene domain. Viewpoints are highlighted as green rectangles. Positions of CTCF-bound and CTCF-less DNase hypersensitive sites outside viewpoints are depicted as blue and orange circles. Bottom: Heatmaps and averaged profiles of DNase sensitivity, CTCF ChIP, H3K4me3 ChIP, merged MChIP-C, merged PLAC-seq and merged promoter-anchored Micro-C signals centered on distal CTCF-bound and CTCF-less DNase hypersensitive sites.

The online version of this article includes the following figure supplement(s) for figure 2:

**Figure supplement 1.** Comparison of MChIP-C with other C-methods.

**Figure supplement 2.** Consensus, PLAC-seq-specific and Micro-C-specific promoter-DHS and promoter-CTCF interactions.

Micro-C data also looks cleaner than PLAC-seq, it still misses some interactions which are obvious from the MChIP-C data.

Next, we compared MChIP-C, PLAC-seq and Micro-C in a more systematic way by examining their merged interaction profiles around promoter-distal DHSs (*Figure 2b*). First, we observed that in MChIP-C 20.6% of PIRs colocalize with CTCF-bound sites (*Figure 1c*; *Supplementary file 1*). We thus separately examined DHSs with and without CTCF, and found that both types of DHSs show increased interaction signal relative to background. To quantify this, we compared the MChIP-C signal at DHS centers to MChIP-C "background" signal within 1–5 kb of the DHS (*Figure 2—figure supplement 1b*), and found a 2.66 median fold enrichment of DHS centers relative to 5 kb background (3.23 with CTCF, and 2.07 without CTCF). In contrast, we found that PLAC-seq shows a very weak enrichment of interactions at DHSs, even for those bound by CTCF (1.25 median fold enrichment with CTCF and 1.33 without CTCF). Although better than PLAC-Seq, Micro-C shows a reduced signal-to-noise ratio at DHSs compared to MChIP-C (1.51 median fold enrichment with CTCF and 1.37 without CTCF). We also compared MChIP-C, PLAC-Seq and Micro-C interaction signal at promoter interacting CTCF and DHS sites, separated into consensus sites (called by all three methods) and method-specific sites (called by only one method; *Figure 2—figure supplement 2*). Remarkably, MChIP-C shows better

sensitivity and resolution not only on the consensus sites but also on sites which we failed to call, suggesting that additional interactions could be discoverable by improving the interaction-calling strategy. Importantly, we note that these aggregate analyses of method sensitivity and resolution should not be affected by differences in sequencing depths.

Thus, in line with other recent reports on MNase-based C-methods (*Hsieh et al., 2020*; *Krietenstein et al., 2020*; *Hua et al., 2021*; *Aljahani et al., 2022*; *Goel et al., 2022*; *Ramasamy et al., 2022*), our results suggest that MChIP-C achieves superior sensitivity and resolution compared to C-methods based on standard restriction enzymes. Additionally, in the context of promoter-centered interactions, our results suggest that MChIP-C also outperforms standard whole-genome Micro-C.

## CTCF orientation-biased interaction of CTCF-bound sites and promoters

As we observed an abundance of CTCF-bound PIRs in MChIP-C data, we asked whether these CTCF-promoter interactions correspond to loop extrusion-driven loops between convergent CTCF sites easily noticeable on the Hi-C heatmaps (*Rao et al., 2014*; *Sanborn et al., 2015*; *Fudenberg et al., 2016*; *Rao et al., 2017*). About half of the identified promoter-CTCF interactions (19,773 out of 38,271) showed CTCF binding at the promoter side as well. Interestingly, only 540 (~2.7%) of those directly correspond to CTCF-CTCF loops reported in a previous Hi-C study (*Rao et al., 2014*). The other half of the MChIP-C interactions containing a CTCF-bound PIR (18,498/38,271) do not show CTCF binding at the promoter side at all. For this set of CTCF-less promoters, we analyzed the orientation of CTCF motifs in PIRs relative to the position of the promoter. We found that promoters preferentially interact with CTCF-bound sites if CTCF motifs are oriented towards the promoter (79% towards vs 21% away) (*Figure 3a*). This bias holds true regardless of the position of the CTCF site relative to the transcription direction (80% vs 20% for upstream CTCF sites and 79% vs 21% for downstream sites). We also observed this bias if both the promoter and the PIR have CTCF ChIP-seq signal, but CTCF motifs are not in convergent/divergent orientation (*Figure 3—figure supplement 1*). We conclude that interactions of promoters with CTCF-bound sites are biased by CTCF orientation, regardless of CTCF binding at the promoter side. This orientation bias suggests the possible involvement of loop extrusion.

## Promoter-interacting enhancers bind a distinct and diverse set of protein factors

We next asked whether the sensitivity and resolution of MChIP-C could allow us to examine the molecular underpinnings of promoter-centered interactions which are not CTCF-based. While the question of what factors underlie P-E interactions is fundamental, this type of analysis would be problematic with restriction enzyme-based C-methods due to their inability to precisely map loop anchors to protein binding sites. To determine protein factors associated with PIRs, we first overlaid MChIP-C PIRs with 271 ChIP-seq profiles for K562 cells (*Supplementary file 1*). Most of the analyzed transcription-related factors and histone post-translational modifications are substantially enriched in MChIP-C PIRs (4–25 X; *Figure 3b*; *Supplementary file 1*). Notable exceptions are facultative heterochromatin-associated histone mark H3K27me3 (1137 overlaps, 1953 [SD = 44] expected by chance) and LINE1-binding protein ZNF146 (183 overlaps, 180 [SD = 12] expected by chance). As expected, CTCF (22,943/1679 [SD = 41]), cohesin subunits RAD21 (10,702/413 [SD = 23]) and SMC3 (12,956/576 [SD = 27]) are among the most highly enriched factors.

Rather than separately consider each factor associated with a PIR, we next used hierarchical clustering to partition PIR-overlapping DHSs (i.e. promoter interacting DHSs; N=19,129) into groups according to the factor binding profile of each DHS. We identified four major clusters (*Figure 3c*; *Figure 3—figure supplement 2a*; *Supplementary file 1*). Clusters 1 (N=4447) and 2 (N=4791) were almost universally bound by the structural factors CTCF (98.1% and 93.6% in clusters 1 and 2 correspondingly), cohesin subunits SMC3 (87.5% and 66.9%) and RAD21 (73.6% and 52.9%) as well as zinc finger protein ZNF143 (87.3% and 66.3%). A total of 24,331 interactions with a median distance of ~92 kb were anchored in DHSs from these two clusters (*Figure 3—figure supplement 2b*). DHSs from clusters 3 (N = 3168) and 4 (N=6723) contained almost no CTCF- or cohesin-bound sites but they were enriched in dozens of transcription-related factors. Notably, CRISPRi-confirmed enhancers (*Fulco et al., 2019*; *Gasperini et al., 2019*) were significantly overrepresented in cluster 3 DHSs

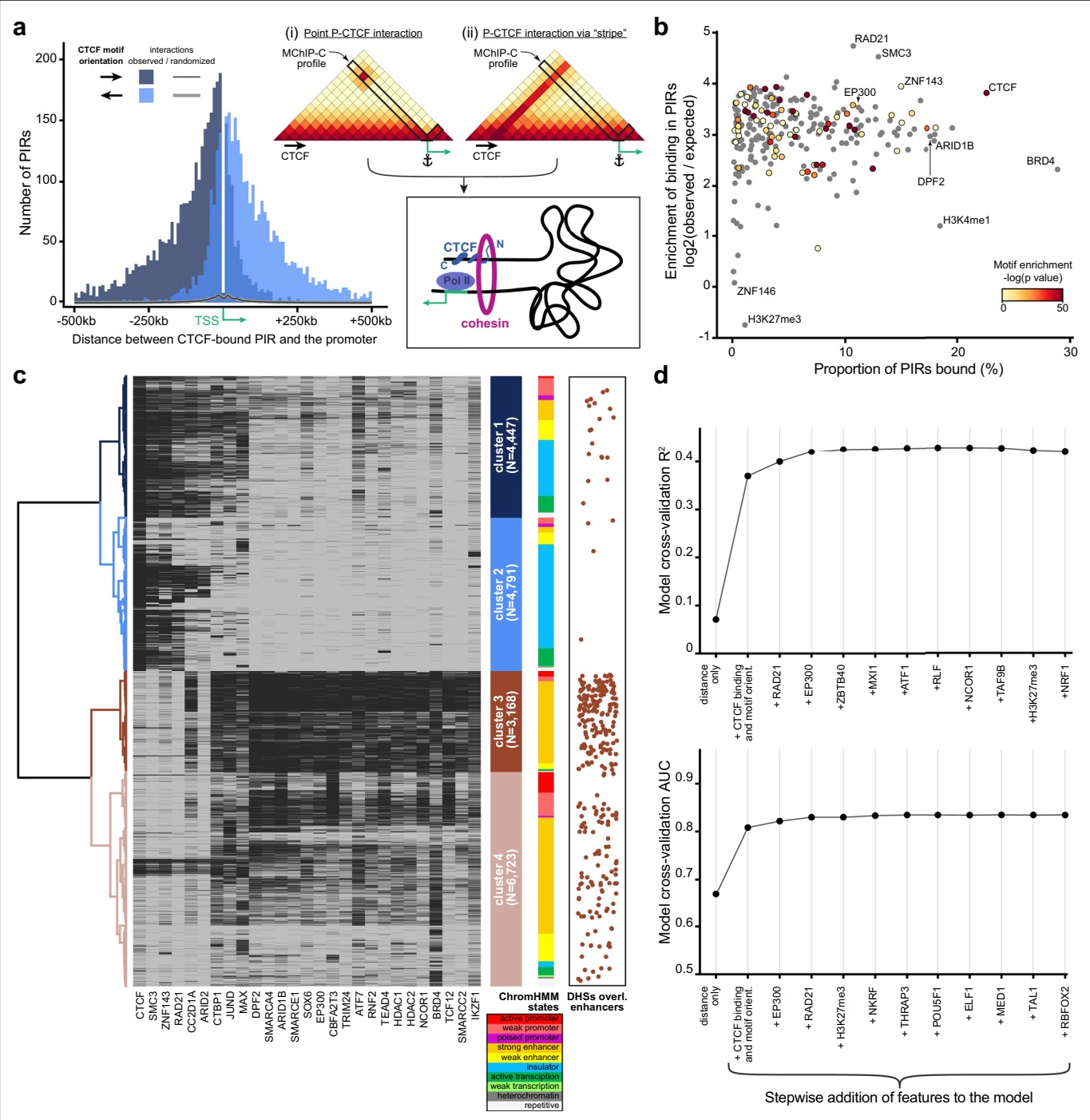

**Figure 3.** Analysis of protein factors underlying MChIP-C interactions. (**a**) Left: CTCF-motif orientation bias in regions interacting with CTCF-less promoters. The majority (~79%) of CTCF motifs are oriented towards the interacting promoter. Right: Schematic of two hypothetical loop extrusion dependent mechanisms that can account for the observed pattern: promoter LE-barrier activity (**i**) or CTCF-originating interaction stripes (**ii**). (**b**) Enrichment of transcription-related factor (TRF) binding in MChIP-C PIRs. Y-axis represents enrichment (log2 observed/expected) of binding for 271 examined TRFs, x-axis – proportion of TRF-bound PIRs, color – enrichment of corresponding motifs in PIRs (grey color is assigned to TRFs lacking DNA-binding motif). (**c**) Hierarchical clustering of PIR-overlapping DHSs (N=19,129). The binding status of 164 TRFs highly enriched in PIRs are used as binary features. Binding of 27 selected TRFs (see Methods) in each PIR-overlapping DHS is shown as a heatmap. ChromHMM chromatin state distributions in each cluster are shown. DHSs overlapping CRISPRi-verified K562 enhancers (*Fulco et al., 2019*; *Gasperini et al., 2019*) are shown as orange dots. (**d**) Predictive performance (3-fold cross-validation R²/AUC) of random forest models predicting MChIP-C signal for DHS-promoter pairs. Starting with an initial model based on distance and CTCF, the most predictive TRF features are added incrementally to the model (left to right).

*Figure 3 continued on next page*

*Figure 3 continued*

The online version of this article includes the following figure supplement(s) for figure 3:

**Figure supplement 1.** CTCF-orientation bias in PIRs interacting with CTCF-occupied promoters.

**Figure supplement 2.** Extended characterization of protein factors underlying promoter-interacting DHSs.

(171/297 overlaps; one-sided binomial test, p=1.64*10$^{-57}$), and we thus labeled P-cluster 3 DHS interactions as regulatory interactions. Overall, there were 7214 regulatory interactions, which tended to be shorter (median length ~47 kb) than structural ones (*Figure 3—figure supplement 2b*). Interestingly, cohesin subunits were absent in the vast majority of class 3 DHSs. In agreement with a number of recent observations (*Thiecke et al., 2020*; *Hsieh et al., 2022*) this may indicate that cohesin is not involved in the maintenance of the regulatory interactions, albeit such a role was suggested earlier (*Kagey et al., 2010*; *Phillips-Cremins et al., 2013*). In conclusion, we find diverse types of promoter interacting DHSs associated with the binding of different sets of factors. However, the clustering analysis was not able to pinpoint a specific factor, beyond CTCF and cohesin, which would parsimoniously explain the observed promoter-DHS interactions.

## Genomic distance, CTCF, cohesin and EP300 are key determinants of promoter-centered interactions

We next sought to pinpoint additional factors which underlie promoter-centered interactions and to directly test their predictive power. We used a greedy forward feature selection approach, based on a random forest regression model trained to predict the strength of MChIP-C interactions between promoters and DHS sites (N=107,570). The initial random forest model consisted of five features: DHS-promoter genomic distance, CTCF ChIP-seq signal at both the promoter and the DHS, and motif orientations on both the promoter and the DHS. This initial model was able to explain ~37% of variance in the MChIP-C signal. Then, we iteratively added the most predictive feature out of a set of 270 ChIP-seq profiles, where predictivity is calculated as R$^2$ in threefold cross-validation. Using this strategy, cohesin subunit RAD21 and histone acetyltransferase EP300 were automatically selected, after which predictive performance plateaued at an R$^2$ of ~43% (*Figure 3d*). We also found the same factors as the most predictive of binary outcomes (high/low MChIP-C signal) reaching an AUC of 0.825.

As CTCF and RAD21 are mostly associated with structural loops, we asked whether EP300 mainly underlies regulatory interactions. Indeed, we find that EP300 was threefold enriched in cluster 3 DHSs and 92.2% of cluster 3 DHSs showed EP300 binding. Interestingly, we noticed that the only proteins that had more binding sites in cluster 3 DHSs are subunits of the SWI/SNF remodeling complex: DPF2 (97.8%), ARID1B (95.6%), and SMARCE1 (94.3%; *Figure 3c*). EP300 has been found to directly interact with SWI/SNF-complex (*Alver et al., 2017*; *Blümli et al., 2021*) and their chromatin binding profiles are highly correlated (Pearson correlation 0.65–0.70). Revisiting the binding profiles of these SWI/SNF subunits, we found their predictive power to be very close to that of EP300 (*Figure 3—figure supplement 2c*). We also specifically examined the predictive power of RNA polymerase II, mediator complex, YY1 and BRD4, which were previously suggested to be associated with enhancer-promoter interaction (*Papantonis and Cook, 2011*; *Kagey et al., 2010*; *Weintraub et al., 2017*; *Hnisz et al., 2017*), and find that while they individually have some predictive power, they are weaker predictors of MChIP-C signal than EP300 and the SWI/SNF subunits (*Figure 3—figure supplement 2c*). Thus, we suggest that EP300 and/or the SWI/SNF complex might be involved in the formation of regulatory chromatin interactions independently of CTCF and cohesin.

## MChIP-C data are largely consistent with the looping model of enhancer activity

The apparent lack of interaction between experimentally-validated enhancers and their cognate promoters in some studies employing C-methods has raised doubts regarding the classical promoter-enhancer looping model. We thus asked whether the enhanced sensitivity and resolution of MChIP-C could shed some light on the fundamental question of whether enhancers should interact with their targets in order to activate them. To address this, we systematically compared MChIP-C profiles with functionally verified E-P pairs identified by CRISPRi screens in K562 (*Fulco et al., 2019*; *Gasperini*

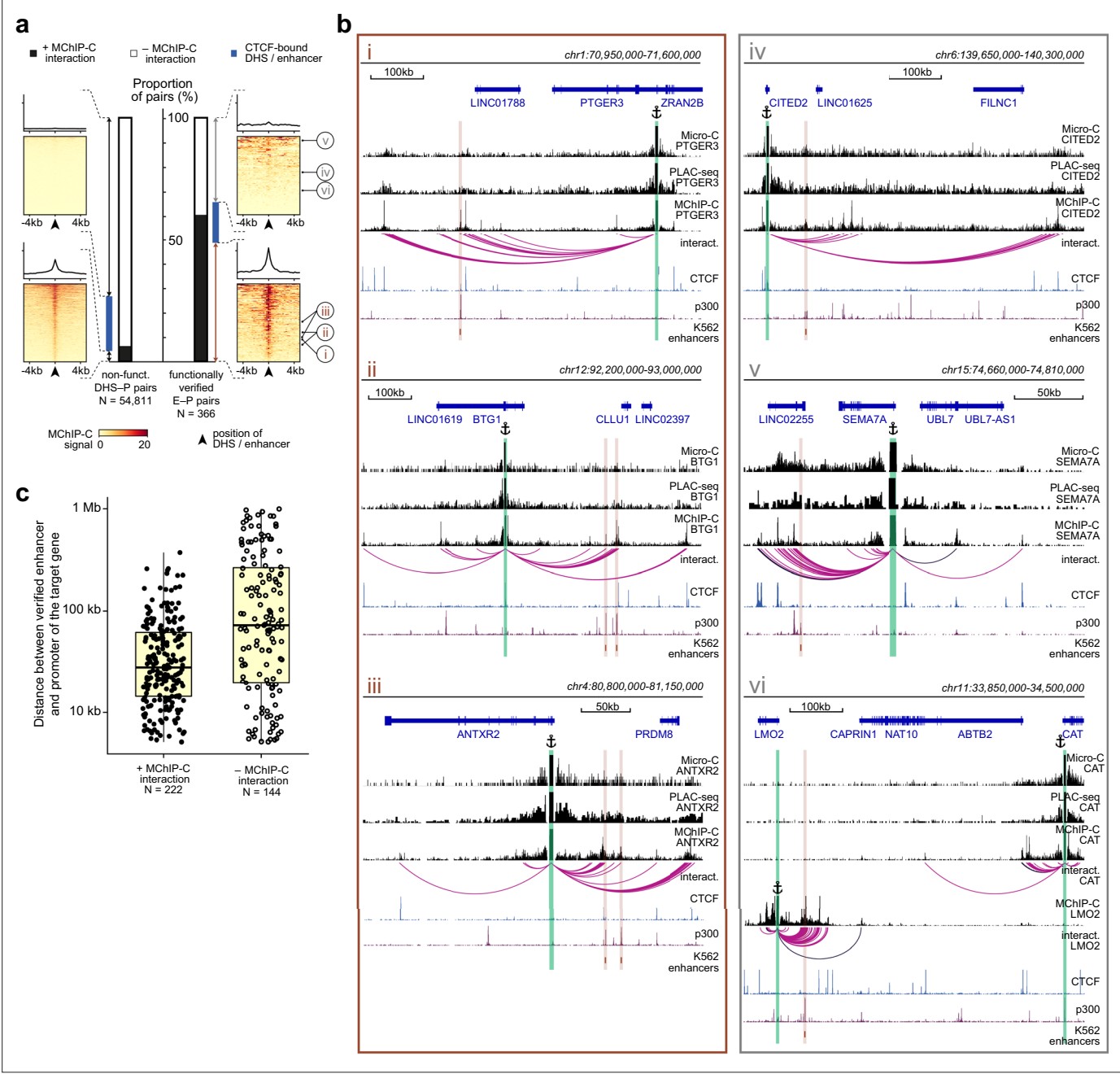

**Figure 4.** The majority of functionally-verified enhancers do physically interact with their target promoters. (a) Bar plots representing the proportion of MChIP-C interacting pairs among nonfunctional DHS-P pairs and CRISPRi-verified E-P pairs. Heatmaps and average profiles of MChIP-C signal are shown for individual subsets. CRISPRi-verified enhancers shown in panel b are indicated by roman numerals. (**b**) Examples of MChIP-C, Micro-C and PLAC-seq profiles for promoters physically interacting with their functionally verified enhancers (i.e. MChIP-C interaction has been found) (i-iii) and not interacting with them (i.e. MChIP-C interaction has not been found) (iv-vi). Viewpoints are highlighted by anchor symbols and green rectangles, enhancers are highlighted by orange rectangles. (**c**) Distance distribution boxplots for verified E-P pairs with and without MChIP-C interactions.

The online version of this article includes the following figure supplement(s) for figure 4:

**Figure supplement 1.** Recall (sensitivity), precision and false positive rate for predictions of functional enhancer-promoter pairs using different C-methods.

*et al., 2019*). We compared 366 functionally verified E-P pairs, in which targeting CRISPRi to the enhancer changed the expression of the target, with 54,811 putative non-regulatory DHS-P pairs, in which CRISPRi showed no effect. Notably, we found that 60.7% (222/366, 95% CI [55.7%; 65.7%]) of the verified pairs have underlying spatial interactions detected with MChIP-C, while only 6.5% (3,554/54,811, 95% CI [6.3%; 6.7%]) of non-regulatory DHS-P pairs show such interactions (*Figure 4a and b*; *Supplementary file 1*). Thus, we conclude that a majority of experimentally validated enhancers exhibit interaction with their target promoters.

Next, we attempted to reproduce this analysis with PLAC-seq, Micro-C, and Hi-C as an additional evaluation of their power to detect enhancer-promoter interactions (*Figure 4—figure supplement 1a and b*; *Supplementary file 1*). We used PLAC-seq interactions reported previously by *Chen et al., 2022*, and for Hi-C and Micro-C we used the entire genome-wide datasets to identify interactions using Mustache (*Roayaei Ardakany et al., 2020*). With respect to recall/sensitivity, the recall of MChIP-C (60.7%, as mentioned above) was superior to those of PLAC-seq (14.0%), Micro-C (19.9%), and Hi-C (2.2%). In spite of this, the precision of the methods was comparable, with 5.9% for MChIP-C, 4.0% for PLAC-seq, 7.4% for Micro-C, and 2.4% for Hi-C. In terms of the false positive rate, MChIP-C was the highest with 6.5%, compared to 1.9% for PLAC-seq, 1.7% for Micro-C, and 0.6% for Hi-C. To control for sequencing depth, we also repeated the analysis with our data down-sampled to 50% so that the valid MChIP-C reads approximately matched the number of valid PLAC-seq reads and the respective number in Micro-C (see Methods). We also further down-sampled the MChIP-C data to 25% and 10% (approximately the amount of total sequenced reads in PLAC-seq). We first observe that although the down-sampling suggests our MChIP-C data is not yet saturated by sequencing depth (*Figure 4—figure supplement 1c*), 50% downsampling still maintains 76.3% of the detected interactions (*Figure 4—figure supplement 1d*). Evaluating predictive performance after down-sampling, we find that 50% down-sampled MChIP-C maintains a high recall of 56.1%, with slightly better precision (6.3%) and false positive rate (5.5%). Even at 10% down-sampling, MChIP-C achieves a better precision (10.4%) and false positive rate (1%) than the competing methods, while its recall (18.2%) is slightly worse than that of Micro-C but better than those of PLAC-Seq and Hi-C. In summary, these results suggest that the enhanced sensitivity of MChIP-C enables detection of a larger and more precise set of enhancer-promoter interactions than the alternatives.

Finally, we inspected more closely cases in which validated E-P pairs did not show interaction in MChIP-C (*Figure 4b*, iv-vi). First, we noticed that, in some instances, weak interactions were visually apparent but were missed by the interaction calling algorithm, especially in viewpoints that had low coverage (*Figure 4b*, iv). Indeed, when we considered 185/366 validated E-P pairs corresponding to high-coverage viewpoints, 68.1% (126/185, 95% CI [61.4%; 74.8%]) had underlying MChIP-C spatial interactions. Second, in some cases we noticed adjacent (few kb) H3K27ac-positive regions clearly interacting with the expected promoter, possibly suggesting inexact identification of the enhancer position due to the limited resolution of the CRISPRi screens (*Gasperini et al., 2019*; *Figure 4b, v*). Third, CRISPRi-detected E-P interactions are not guaranteed to represent direct effects. In some cases, an enhancer may impose secondary influence through the action of the directly targeted gene (*Gasperini et al., 2019*). For example, CRISPRi showed that an enhancer near LMO2 affects both LMO2 and CAT, a gene approximately 400 kb away. MChIP-C only shows interaction with LMO2 but not with CAT (*Figure 4b, vi*). We hypothesize that regulatory relationships between this enhancer and CAT are mediated through transcription factor encoded by the LMO2 gene and thus it is unreasonable to expect spatial interaction between them. Additionally, we expect that interactions between enhancers and indirect targets would not be overrepresented at short genomic distances. Indeed, we observed that the genomic distance between non-interacting E-P pairs (median 71 kb) is substantially larger than between interacting E-P pairs (median 30 kb), supporting the possibility of indirect activation (*Figure 4c*). Another possible source of indirect effects are CTCF-occupied insulators. As was pointed out previously (*Fulco et al., 2019*), these sites can affect the expression of genes in CRISPRi screens, but do not need to spatially interact with their targets. Indeed, 59 of 366 verified E-P pairs show CTCF occupancy at enhancer side, and 13 of these lack E-P interaction in MChIP-C. In summary, we conclude that the fraction of verified E-P pairs in which we observe a spatial interaction is likely underestimated, which further supports the notion that most enhancers physically interact with their targets and that many such interactions may have not been identified previously due to technical limitations of the employed methods.

## Discussion

In this study, we present a new high-resolution genome-wide C-method, MChIP-C, which integrates proximity ligation of MNase-digested chromatin with chromatin immunoprecipitation and deep sequencing. Like other MNase-based C-methods, MChIP-C allows precise mapping of localized chromatin interactions including E-P contacts which often evade capture by restriction endonuclease-based C-methods (*Goel and Hansen, 2021*). Uniquely, MChIP-C combines the resolution and sensitivity of MCC with the genome-wide scale of Micro-C.

Applying H3K4me3-directed MChIP-C to K562 cells, we explored the spatial connectivity of active promoters. Single-nucleosome resolution and high sensitivity allowed us to map tens of thousands of localized spatial interactions between promoters and distal genomic sites. We focused on promoter interactions with non-promoter elements (PIRs). Although P-P interactions were clearly visible in our maps, they were excluded from most of the analyses. The promoter-centered map of chromatin spatial contacts represents a valuable resource for further research. The quality of the resulting genome-wide profiles is supported by the fact that previously identified spatial contacts, such as contacts of globin gene promoters with their enhancers, are clearly visible.

Several types of C-methods have been previously used to study enhancer-promoter interactions. Since genome-wide C-methods require very deep sequencing to reach high resolutions, targeted C-methods have been useful in alleviating some of the sequencing burden. However, any selection steps can potentially reduce the complexity of the resulting library. We directly compared MChIP-C interaction data to that of both restriction endonuclease-based (genome-wide Hi-C and targeted PLAC-Seq) methods and MNase-based (genome-wide Micro-C) methods. In order to rigorously assess the sensitivity and resolution of MChIP-C beyond visual evaluation, we performed comparative analyses of both raw data (on CTCF and DHS sites) and called interactions (on CRISPRi-validated E-P pairs). We find that MChIP-C generally outperforms PLAC-seq, Micro-C and Hi-C, resulting in a larger repertoire of promoter-based interactions which also benefit from high resolution and increased signal-to-noise ratio. MChIP-C avoids the incorporation of biotin to enrich ligated fragments, thus yielding highly complex libraries. Consequently, full realization of MChIP-C potential currently requires a significant sequencing depth due to the small fraction of informative reads. We anticipate that further experimental optimizations of MChIP-C could significantly increase the method's yield.

In agreement with previous observations, promoters appear to interact primarily with the surrounding CTCF sites. Interestingly, CTCF-motifs in these sites are typically oriented towards the interacting promoter (*Valton et al., 2022*). This observation can be explained by the putative ability of active promoters to block movement of the loop extrusion complex which can lead to the establishment of promoter-CTCF loops with CTCF motifs preferentially oriented towards the promoters (*Figure 3a, i*; *Banigan et al., 2022*). Alternatively, the observed bias might be a consequence of a more general phenomenon known as architectural stripes, where a CTCF-bound site blocks loop extrusion in a directional manner, causing nonspecific interactions with an extended nearby region (*Vian et al., 2018*; which may include the gene promoter; *Figure 3a, ii*).

Similarly to other MNase-based C-methods, MChIP-C is sensitive enough to identify many relatively weak chromatin interactions that are independent of CTCF binding. The most intriguing group of such spatial contacts is E-P regulatory interactions. We identified 7214 interactions linking active promoters with distal DHSs bound by a set of enhancer-specific factors. These DHSs are highly enriched with functionally verified K562 regulatory sequences and contain chromatin marks typical for active enhancers (*Figure 3c*; *Figure 3—figure supplement 2a*). The increased proximity between promoters and these enhancer-like DHSs presumably represents regulatory E-P interactions.

One of the long-standing hypotheses explaining enhancer action at a distance is the E-P looping model (*Popay and Dixon, 2022*; *Ptashne, 1986*) which assumes that direct physical contact between an enhancer and its target promoter is an essential prerequisite for the promoter activation. Although it is known that at least some enhancers do establish physical contacts with their targets, technical limitations of methods assessing spatial proximity as well as scarcity of functional data on mammalian enhancers prevented accurate systematic evaluation of the E-P looping model. To explain the elusiveness of E-P interactions, it has been suggested that their highly dynamic nature may preclude capturing with existing proximity ligation techniques (*Schoenfelder and Fraser, 2019*). Here, we showed that 60.7% of functional enhancer-promoter pairs previously identified in K562 cells using CRISPRi screens (*Fulco et al., 2019*; *Gasperini et al., 2019*) do establish interactions, implying

their spatial proximity. Moreover, due to various technical reasons, including insufficient MChIP-C coverage, CRISPRi enhancer mapping mistakes, and indirect regulatory effects, this fraction probably represents an underestimation of the true E-P interaction rate. Thus, it is likely that the majority of active enhancers directly interact with target promoters.

In addition to identifying interactions between promoters and their CRISPRi-validated enhancers, MChIP-C found many more apparently non-functional interactions of promoters with DHSs which did not show a significant effect on gene expression. Such interactions are also often observed in alternative C-methods, and in some cases could represent CRISPRi false negative pairs. These interactions could also include structural loops as well as pre-established interactions of promoters with conditionally-activated regulatory elements (*Jin et al., 2013*; *Comoglio et al., 2018*), and warrant further investigation leveraging the increased sensitivity and resolution of MChIP-C.

Although it is generally assumed that in the C-methods ligation occurs only between fragments that are in close spatial proximity, the mobility of DNA ends within fixed nuclei has not been extensively studied. It is possible that DNA ends available for ligation can scan a certain territory and thus MChIP-C detected interactions do not exactly represent direct physical contacts. However, we argue that our data demonstrates that, in most cases, enhancers establish some form of spatial interaction with their targets, resulting in increased ligation frequency.

Although we have shown that active mammalian enhancers in general interact with the target promoters, we cannot infer that this interaction is necessary for target activation. Testing whether it is possible to decouple spatial interactions from gene activity would require understanding the physical nature of these interactions. Various protein factors and a number of physical mechanisms including canonical protein-protein interactions, molecular crowding and liquid-liquid phase separation were implicated in this phenomenon (*Kagey et al., 2010*; *Papantonis and Cook, 2013*; *Hnisz et al., 2017*). One protein complex which was suggested to be involved is cohesin (*Kagey et al., 2010*; *Phillips-Cremins et al., 2013*). However, in line with studies exploring cellular responses to the acute loss of the cohesin complex (*Thiecke et al., 2020*; *Hsieh et al., 2022*), we observed almost no overlap between regulatory E-P interactions and cohesin binding. Different groups proposed other candidates for the role: the mediator complex, BRD4, YY1 and RNA polymerase II (*Kagey et al., 2010*; *Phillips-Cremins et al., 2013*; *Hnisz et al., 2017*; *Weintraub et al., 2017*; *Papantonis and Cook, 2013*). However, recent studies did not identify drastic changes in E-P interaction frequency caused by the degradation or the inhibition of any of these proteins (*Ramasamy et al., 2022*; *Crump et al., 2021*; *Hsieh et al., 2022*; *Zhang et al., 2022*).

In order to identify proteins that may be involved in E-P interaction, we assessed the ability of an assortment of ChIP-seq binding profiles to predict the strength of promoter-PIR interaction measured by MChIP-C. As expected, the binding of CTCF and cohesin best predicted the strength of the MChIP-C signal, as the latter is dominated by structural promoter-CTCF interactions. However, we also identified EP300 and SWI/SNF complex as strong predictors of MChIP-C interaction, and they are enriched in enhancer-like PIRs lacking CTCF. These hallmark enhancer proteins were previously shown to bind each other and to share chromatin interaction sites genome-wide (*Alver et al., 2017*; *Blümli et al., 2021*). Unlike cohesin, mediator, BRD4, YY1 and RNA polymerase II, EP300 and SWI/SNF have rarely been considered as mediators of E-P interaction. However, in our analysis, they consistently outperformed other enhancer-associated factors in terms of predictive power. Although these results may be affected by differences in the quality of the ChIP-seq data, the potential role of EP300 and SWI/SNF in E-P interaction warrants further experimental exploration. Since both EP300 and SWI/SNF are known as chromatin modifiers rather than structural proteins (albeit some studies suggest such a role for EP300 *Ma et al., 2021*), it is possible that their participation in E-P interaction is indirect. To this end, a scenario involving histone acetylation by EP300 may be considered. Although H3K27 residue is a well-documented EP300 nucleosomal target (*Tie et al., 2009*), a recent study identified several other lysine residues localized in H2B N-terminus that are acetylated by EP300 (*Weinert et al., 2018*; *Narita et al., 2022a*; *Narita et al., 2022b*). These H2B N-terminus acetylation marks correlate with the enhancer activity more strongly than H3K27ac and therefore qualify for being involved in E-P interaction.

# Methods

**Key resources table**

| Reagent type (species) or resource | Designation | Source or reference | Identifiers | Additional information |
|---|---|---|---|---|
| Cell line (*Homo sapiens*) | K562 | ATCC | ATCC:CCL-243 | |
| Antibody | anti-H3K4me3 (rabbit polyclonal) | Active Motif | Active Motif:39016 | (1:200) |
| Chemical compound, drug | Digitonin | Sigma-Aldrich | Sigma-Aldrich:D-5628 | |
| Chemical compound, drug | Protease Inhibitor Cocktail | Bimake | Bimake:B14001 | |
| Peptide, recombinant protein | Micrococcal Nuclease | Thermo Fisher Scientific | Thermo Fisher Scientific:EN0181 | |
| Peptide, recombinant protein | T4 Polynucleotide Kinase | New England Biolabs | New England Biolabs:M0201L | |
| Peptide, recombinant protein | DNA Polymerase I, Klenow Fragment | New England Biolabs | New England Biolabs:M0210L | |
| Peptide, recombinant protein | T4 DNA Ligase | Thermo Fisher Scientific | Thermo Fisher Scientific:EL0012 | |
| Commercial assay or kit | Protein A/G Magnetic Beads | Thermo Fisher Scientific | Thermo Fisher Scientific:88802 | |
| Commercial assay or kit | NEBNext Ultra II DNA Library Prep Kit | New England Biolabs | New England Biolabs:E7645 | |
| Commercial assay or kit | TruSeq DNA Single Indexes | Illumina | Illumina: 20015960 and Illumina:20015961 | |
| Commercial assay or kit | KAPA HiFi HotStart PCR Kit | Roche | Roche:07958897001 | |
| Software, algorithm | bwa, v.0.7.17 | *Li, 2013* | RRID:SCR_010910 | https://github.com/lh3/bwa |
| Software, algorithm | Bowtie2, v.2.3.4 | *Langmead and Salzberg, 2012* | RRID:SCR_016368 | https://bowtie-bio.sourceforge.net/bowtie2/index.shtml |
| Software, algorithm | pairtools, v.0.3.0 | *Abdennur et al., 2023* | RRID:SCR_023038 | https://github.com/open2c/pairtools |
| Software, algorithm | samtools, v.1.15.1 | *Danecek et al., 2021* | RRID:SCR_002105 | https://github.com/samtools/samtools |
| Software, algorithm | bedtools, v2.26.0 | *Quinlan and Hall, 2010* | RRID:SCR_006646 | https://github.com/arq5x/bedtools2 |
| Software, algorithm | Python, v.3.7.12 | | RRID:SCR_008394 | https://www.python.org |
| Software, algorithm | numpy, v.1.21.6, | *Harris et al., 2020* | RRID:SCR_008633 | https://github.com/numpy/numpy |
| Software, algorithm | pandas, v.1.3.5 | | RRID:SCR_018214 | https://github.com/pandas-dev/pandas |
| Software, algorithm | matplotlib, v.3.5.3 | *Hunter, 2007* | RRID:SCR_008624 | https://github.com/matplotlib/matplotlib |
| Software, algorithm | cooler, v.0.9.1 | *Abdennur and Mirny, 2020* | RRID:SCR_024194 | https://github.com/open2c/cooler |
| Software, algorithm | cooltools, v.0.5.1 | *Abdennur et al., 2024* | RRID:SCR_026118 | https://github.com/open2c/cooltools |
| Software, algorithm | R, v. 4.2.1 | | RRID:SCR_001905 | https://cran.r-project.org |
| software, algorithm | dplyr, v.1.0.9 | | RRID:SCR_016708 | https://github.com/tidyverse/dplyr |
| Software, algorithm | tidyr, v.1.2.0 | | RRID:SCR_017102 | https://github.com/tidyverse/tidyr |

*Continued on next page*

*Continued*

| Reagent type (species) or resource | Designation | Source or reference | Identifiers | Additional information |
|---|---|---|---|---|
| Software, algorithm | ggplot2, v.3.3.6 | | RRID:SCR_014601 | https://github.com/tidyverse/ggplot2 |
| Software, algorithm | gplots, v. 3.1.3 | | RRID:SCR_025035 | https://github.com/talgalili/gplots |
| Software, algorithm | data.table, v.1.14.8 | | RRID:SCR_026117 | https://github.com/Rdatatable/data.table |
| Software, algorithm | GenomicRanges, v.1.48.0 | *Lawrence et al., 2013* | RRID:SCR_000025 | https://github.com/Bioconductor/GenomicRanges |
| Software, algorithm | reshape2, v.1.4.4 | | RRID:SCR_022679 | https://github.com/cran/reshape2 |
| Software, algorithm | fitdistrplus, v.1.1–8 | *Delignette-Muller and Dutang, 2015* | RRID:SCR_024274 | https://github.com/lbbe-software/fitdistrplus |
| Software, algorithm | RColorBrewer, v.1.1–3 | | RRID:SCR_016697 | https://github.com/cran/RColorBrewer |
| Software, algorithm | dendextend, v.1.17.1 | *Galili, 2015* | RRID:SCR_026116 | https://github.com/talgalili/dendextend |
| Software, algorithm | dendroextras, v.0.2.3 | | RRID:SCR_026115 | https://github.com/jefferis/dendroextras |
| Software, algorithm | GGally, v.2.1.2 | | RRID:SCR_026114 | https://github.com/ggobi/ggally |
| Software, algorithm | gridExtra, v.2.3 | | RRID:SCR_025249 | https://github.com/baptiste/gridExtra |
| Software, algorithm | eulerr, v.7.0.1 | *Larsson and Gustafsson, 2018* | RRID:SCR_022753 | https://github.com/jolars/eulerr |
| Software, algorithm | ranger, v.0.16.0 | *Wright and Ziegler, 2017* | RRID:SCR_022521 | https://github.com/imbs-hl/ranger |
| Software, algorithm | caret, v.6.0–93 | *Kuhn, 2008* | RRID:SCR_022524 | https://github.com/topepo/caret |
| Software, algorithm | PRROC, v.1.3.1 | | RRID:SCR_026113 | https://github.com/cran/PRROC |
| Software, algorithm | UpSetR, v.1.4.0 | *Conway et al., 2017* | RRID:SCR_026112 | https://github.com/hms-dbmi/UpSetR |
| Software, algorithm | HOMER, v.4.11.1 | *Heinz et al., 2010* | RRID:SCR_010881 | http://homer.ucsd.edu/homer/motif/ |
| Software, algorithm | CrossMap, v.0.6.0 | *Zhao et al., 2014* | RRID:SCR_001173 | https://github.com/liguowang/CrossMap |
| Software, algorithm | nextflow, v.22.10.4 | *Di Tommaso et al., 2017* | RRID:SCR_024135 | https://github.com/nextflow-io/nextflow |
| Software, algorithm | ditiller-nf pipeline, v.0.3.4 | | RRID:SCR_026111 | https://github.com/open2c/distiller-nf |
| Software, algorithm | Mustache, v.1.3.2 | *Roayaei Ardakany et al., 2020* | RRID:SCR_026110 | https://github.com/ay-lab/mustache |
| Software, algorithm | bedGraphToBigWig | | | http://hgdownload.soe.ucsc.edu/admin/exe/ |
| Software, algorithm | liftOver | | | http://hgdownload.soe.ucsc.edu/admin/exe/ |
| Software, algorithm | wigToBigWig | | | http://hgdownload.soe.ucsc.edu/admin/exe/ |

*Continued on next page*

*Continued*

| Reagent type (species) or resource | Designation | Source or reference | Identifiers | Additional information |
|---|---|---|---|---|
| Software, algorithm | Integrative Genomics Viewer, v.2.8.0 | *Robinson et al., 2011* | | https://igv.org/doc/desktop/ |
| Software, algorithm | Adobe Illustrator, v.23.0.1 | | | https://www.adobe.com/ products/illustrator.html |

## MChIP-C experimental procedure

K562 cells (ATCC) were cultivated in DMEM medium supplemented with 10% fetal bovine serum and 1×penicillin/streptomycin in a humidified 37 °C incubator with 5% $CO_2$. ~3.5 M cells were crosslinked for 10 min at room temperature in 2.5 mL of fresh full growth medium supplemented with 2% formaldehyde (Sigma-Aldrich, F8775). To quench the reaction, glycine was added to reach a final concentration of 0.2 M; the suspension was immediately transferred to ice and incubated there for 5 min. Crosslinked cells were centrifuged at 300 × *g* and 4 °C, washed with cold phosphate-buffered saline (PBS) and centrifuged again. Pellets were resuspended in 350 µL of cold PBS supplemented with 0.5×protease inhibitor cocktail (PIC; Bimake, B14001) and 0.5 mM PMSF. Digitonin (Sigma-Aldrich, D-5628) stock solution (1% in DMSO) was added to cells to reach a final concentration of 0.01%. A 1 mL tip was used to carefully mix the suspension. Cells were permeabilized on ice for 7 min then centrifuged at 300 × *g* and 4 °C and resuspended in 800 µL of MNase digestion buffer (10 mM Tris-HCl pH 7.5, 1 mM $CaCl_2$). To achieve a sufficient level of chromatin digestion (50–75% of DNA in mononucleosomal fragments) we added 35–40 U of MNase (Thermo Scientific, EN0181) and incubated the cells at 37 °C with shaking for 1 h. EGTA was added to reach a final concentration of 5 mM in order to stop digestion. Cells were centrifuged at 300 × *g* and room temperature, resuspended with a low-retention tip in PBS and transferred to a new, low-retention, tube. We pelleted the material once again at 1000 × *g* and resuspended it in 530 µL 1.05×ligation buffer (ThermoScientific, EL0012) supplemented with 0.105 mM dNTPs and 2.1 mM ATP. 50 µL of the material at this stage was separated as a control of digestion efficiency. 100 U of T4 Polynucleotide Kinase (NEB, M0201L) and 50 U of Klenow fragment of DNA I polymerase (NEB, M0210L) were added to the remaining material in order to repair DNA ends; the mix was incubated at 37 °C with shaking. After 1 hr 37.5 U of T4 DNA ligase (ThermoScientific, EL0012) were added to the reaction and incubation was continued at 37 °C for an additional hour. Before leaving the ligation reaction overnight, the mixer was cooled to 20 °C and 37.5 U more of T4 DNA ligase were added.

The next day, we took 50 µL aliquot as a ligation control, transferred the remaining chromatin to a 2 mL tube and centrifuged it at 5000 × *g*. The pellet was resuspended in 550 µL of ice cold ChIP lysis buffer (50 mM Tris-HCl pH 8.0, 1% SDS, 10 mM EDTA) supplemented with 0.5×PIC (Bimake, B14001) and 0.5 mM PMSF. All buffers used in the following sonication and immunoprecipitation steps were precooled to 4 °C. We solubilized chromatin with 3 15 s ultrasound pulses on '15' power setting of VirSonic 100 (VirTis) sonicator. Then, we cleared the solubilized chromatin with 21,000 × *g* centrifugation and transferred the supernatant fraction to a 30 kDa Amicon filter (Millipore, UFC503096). We changed the solvent with two successive washes with RIPA buffer (50 mM Tris–HCl pH 8.0, 150 mM NaCl, 1% Triton X-100 (v/v), 0.5% sodium deoxycholate, 0.1% SDS). After the final wash, we transferred the material to a new 1.5 mL tube, brought the volume of RIPA to 1 mL and supplemented it with 1×PIC (Bimake, B14001). Subsequent incubation with antibodies, immunoprecipitation and DNA isolation were performed as previously described (*Golov et al., 2021*). We used 5 µg of polyclonal anti-H3K4me3 antibodies (Active motif, 39016) per MChIP-C experiment. DNA of digestion and ligation controls was isolated in parallel with the immunoprecipitated DNA. After ethanol precipitation, DNA was reconstituted from each sample (controls and the MChIP-C) in 10 µL of 10 mM Tris-HCl buffer (pH 8.0). To assess efficiency of the digestion and ligation, the control samples were run on a 1.5% agarose gel (*Figure 1—figure supplement 1a*).

MChIP-C NGS libraries were prepared from immunoprecipitated DNA with the NEBNext Ultra II DNA Library Prep Kit for Illumina (NEB, E7645) according to the manufacturer's recommendations, with the following modifications: (1) we decreased the reaction volumes two-fold; (2) instead of the adaptor included in the kit, we used 1 µL of barcoded Y-shaped Illumina TruSeq adapters (Illumina, 20015960 and 20015961) per reaction and extended ligation time to 1 hr. We used 17 µL (~0.37×)

of AMPure XP beads (Beckman Coulter, A63881) in the pre-PCR clean-up to select longer DNA fragments and therefore enrich the final library with ligation-informative pairs. Adapter-ligated material was amplified with 12–14 cycles of PCR using KAPA HiFi HotStart PCR Kit (Roche, 07958897001) and universal P5/P7 Illumina primers (see *Golov et al., 2020*). Post-PCR clean-up was performed without size selection using 1.3×volume of AMPure XP beads (Beckman Coulter, A63881).

Overall, we performed 4 biological replicates of H3K4me3 MChIP-C. The libraries were paired-end sequenced (PE100) on Illumina NovaSeq 6000 and BGI DNBSEQ-T7 devices with ~0.5–1.5 B read pairs per library (*Figure 1—figure supplement 1b*, *Supplementary file 1*).

## H3K4me3 ChIP-seq

~3.5M K562 cells were crosslinked and permeabilized as described above. Permeabilized cells were resuspended in 550 µL of ice cold ChIP lysis buffer (50 mM Tris-HCl pH 8.0, 1% SDS, 10 mM EDTA) supplemented with 0.5×PIC (Bimake, B14001) and 0.5 mM PMSF. Chromatin was sheared with 4 30 s ultrasound pulses on '15' power setting of VirSonic 100 (VirTis) sonicator. Incubation with antibodies, immunoprecipitation, DNA isolation and NGS library preparation were performed as described MChIP-C protocol, except for the pre-PCR clean-up step where DNA was cleaned with 35 µL (0.76×) of AMPure XP beads (Beckman Coulter, A63881). 12 cycles of PCR were used to amplify the ChIP-seq library, and it was paired-end sequenced (PE100) on an Illumina NovaSeq 6000 sequencer with ~6 M read pairs. The sequencing reads were aligned to the hg19 genome build and a genome-wide H3K4me3 occupancy profile was generated as described previously (*Golov et al., 2021*).

## MChIP-C data primary processing

The sequenced reads of each replicate were mapped to the hg19 genome build with BWA-MEM (with -SP5M flags) (*Li, 2013*) and parsed with the pairtools package (v.0.3.0; *Abdennur et al., 2023*). Mononucleosomal (>100 bp and <201 bp apart; convergent) and distal (>5000 bp apart; cis) read pairs were isolated for downstream analysis.

Mononucleosomal reads were down-sampled to 20 M pairs, deduplicated with pairtools and converted to BAM files using samtools (v.1.15.1) (*Danecek et al., 2021*). Mononucleosomal peaks for each individual replicate were called with MACS2 (v.2.2.7.1; FDR <0.0001, maximum gap = 1000 bp) (*Zhang et al., 2008*) and 50 bp occupancy profiles were generated with deeptools (v.3.5.1; *Ramírez et al., 2016*). To compare mononucleosomal profiles of replicates with each other and with conventional H3K4me3 ChIP-seq signal, we calculated read coverage in K562 DHSs (see *Supplementary file 1* for the source of K562 DHS data) for each experiment with deeptools ('multiBigwigSummary BEDfile' command), and compared pairwise Pearson correlation coefficients (*Figure 1—figure supplement 1d*).

We considered genomic regions overlapping mononucleosomal peaks in at least 3 replicates to be consensus peaks. Nearby (<1000 bp apart) consensus peaks were merged with bedtools (v2.26.0; *Quinlan and Hall, 2010*) resulting in 11,743 MChIP-C viewpoints (*Supplementary file 1*). DNase sensitivity, H3K4me3 ChIP and CAGE signals (see *Supplementary file 1* for the source of K562 CAGE data, two CAGE replicates were averaged with bigwigCompare from deeptools) in a 10 kb window centered on each MChIP-C viewpoint and distal DHSs were calculated and plotted with deeptools ('computeMatrix reference-point' and 'plotHeatmap' commands; *Figure 1—figure supplement 1e*).

Next, we deduplicated distal read pairs using pairtools (*Abdennur et al., 2023*), produced BAM files and filtered out read pairs that did not map to the identified MChIP-C viewpoints using samtools. Filtered binary files were converted to replicate-specific coverage profiles, or merged MChIP-C profiles, which were then piled up together into an aggregate merged profile. To analyze and visualize interaction profiles of individual viewpoints, we selected from the merged profile a subset of reads mapping to the viewpoint of interest with samtools ('samtools view -P' command). To plot merged and viewpoint specific MChIP-C interaction profiles the data was binned in 100 bp or 250 bp bins.

To evaluate reproducibility of the distal MChIP-C signal in replicates and to identify promoter-centered interactions, we analyzed the distribution of MChIP-C reads in 250 bp genomic bins. We started with deduplicated distal read pairsam files produced by pairtools; using the data on MChIP-C viewpoints, we extracted positions of distal read pairs mapping to the viewpoints with pairtools ('pairtools split' command) and bedtools ('bedtools pairtobed' command). Each of these read pairs was assigned to a viewpoint; if two reads in a pair mapped to two different viewpoints,

the pair was assigned to both viewpoints. Then we overlapped ('bedtools intersect' command) the positions of the mate reads (other-end reads) with 250 bp genomic bins and aggregated the pairs mapping to the same viewpoint and the same other-end bin generating four replicate-specific BEDPE files. Each line of the resulting files contained the count of read pairs for a unique viewpoint-genomic bin (other-end bin) pair. To calculate inter-replicate correlation of viewpoint-separated MChIP-C signal, we selected pairs contained within a 1 Mb genomic window with only one end mapping inside a viewpoint and estimated inter-replicate Pearson correlation (*Figure 1—figure supplement 1f*). We also estimated inter-replicate correlation of the merged data. To achieve this, we used merged MChIP-C profiles to create 250 bp-binned coverage dataframes with deeptools ('multiBigwigSummary bins' command), then filtered out bins overlapping viewpoints as well as all bins containing zero reads in at least one replicate and finally calculated inter-replicate Pearson correlation coefficients (*Figure 1—figure supplement 1f*). To generate distance decay curves, we calculated aggregated MChIP-C signal from replicate-specific BEDPE files in 30 distance bins of equal size on a log10 scale between 3.5 (3,162 bp) and 6.5 (3,162 kbp; *Figure 1—figure supplement 1g*).

Using cooler (v.0.9.1; *Abdennur and Mirny, 2020*), we converted files containing MChIP-C pairs into replicate-specific mcool files with a minimal resolution of 250 bp. We also pooled the 4 mcool files into a single genome-wide contact matrix.

## MChIP-C interaction calling

Before calling interactions, we normalized MChIP-C signal in each replicate by the replicate-matched mononucleosomal signal. We did this to account for differences of genomic regions in their propensity to be immunoprecipitated with H3K4me3 antibodies. To consider both viewpoint and other-end bin H3K4me3 occupancy, we normalized by mean mononucleosomal coverage of these two regions (we used 'samtools bedcov' command to estimate coverage of each viewpoint and each genomic bin).

We first identified significantly interacting viewpoint-other-end pairs in each replicate separately. To reduce noise and avoid calling false positive interactions, we focused on 10,949 viewpoints with a raw total coverage (viewpoint coverage in all four replicates combined) of more than 1000 read pairs. In order to perform interaction calling, we wanted different viewpoint-specific profiles to be comparable, thus we filtered out pairs separated by more than 2.5 Mb and normalized signal once again, by viewpoint coverage. The viewpoint coverage was defined as a sum of all H3K4me3 occupancy-normalized MChIP-C signal pertaining to the specific viewpoint. The final MChIP-C signal was normalized twice (by H3K4me3 occupancy and by viewpoint coverage), we refer to it as normalized signal. To identify viewpoint- other-end pairs with signal significantly exceeding the background, we first assigned each analyzed pair with the distance rank (from 1 for other-end bins directly neighboring the viewpoints to 10,000 for bins separated by 2.5 Mb from the viewpoints). Then, in each distance rank starting with the 20th (since pairs of distal reads were separated by at least 5000 bp we had almost no signal in the first 19 distance ranks) and ending with the 4000th, we fitted a Weibull distribution-based background model and called pairs with p-value <0.01 and more than three reads interactions. We also included pairs with distance rank more than 4000 and more than three reads in replicate-specific interaction lists.

We combined all viewpoint-other-end pairs identified as significant interactions in at least one replicate in one dataset. We tested whether each of the interactions in this combined list had a corresponding viewpoint-other-end pair in each of the four individual replicates (we defined correspondence as cases where viewpoints are exactly aligned and other ends are located within 1 kb from each other). Then for each replicate we calculated the proportion of interactions without a corresponding interaction present in at least one another replicate.

Next, we identified significantly interacting viewpoint-other-end pairs in an aggregate MChIP-C dataset. We summarized all replicate-specific distal MChIP-C signals in each viewpoint-other-end pair and performed the interaction-calling procedure described above with a minor modification: we increased to 6 the threshold for minimal number of raw reads in bins which we considered to call viewpoint interacting bins. Finally, we divided all identified interactions into two groups: promoter-promoter (P-P) and promoter-nonpromoter (P-PIR) – depending on whether the other-end bin was localized within a viewpoint (*Supplementary file 1*; *Figure 1c*).

## Analysis of MChIP-C interactions crossing TAD boundary

We used K562 TAD boundaries from *Rao et al., 2014*. The expected number of interactions crossing TAD boundaries was defined as a mean count of cross-boundary interactions in 100 independent random permutations changing the positions of the interaction other-ends while preserving the interaction distance distribution. The position of other-end bins in each permutation was established by random reshuffling of distances separating viewpoints from their original interaction partners (bins within other viewpoints or PIRs).

## MCC data reanalysis

To estimate the typical proportion of valid ligation pairs in MCC sequencing output, we reanalyzed publicly available datasets from *Hua et al., 2021* and *Downes et al., 2021*. We downloaded fastq files for at least three biological replicates from each interrogated cell type and mapped them to hg19 or mm10 genomes with bwa (bwa mem -SP5M). Then we parsed reads with pairtools and calculated the fraction of cis-read pairs separated by more than 5000 bp.

## PLAC-seq data reanalysis

To compare MChIP-C data with a similar restriction-endonuclease based C-method, we reanalyzed publicly available H3K4me3 PLAC-seq experiments (*Chen et al., 2022*). Raw sequencing reads were downloaded from the 4D Nucleome Data portal (here) and processed similarly to MChIP-C reads. We relied on MChIP-C-defined viewpoints to obtain merged PLAC-seq profiles and viewpoint-specific PLAC-seq profiles. We used restriction fragment-length bins to build viewpoint-specific PLAC-seq profiles and fixed 250 bp bins to build merged profiles.

For downstream analysis, we utilized PLAC-seq interactions identified by the authors of the original publication (GEO, GSE161873). We merged interactions identified in two individual experimental replicates in one set and lifted them to hg19 genome assembly with the UCSC liftOver tool.

## Micro-C data reanalysis

We also compared MChIP-C with a recently published K562 Micro-C data (*Barshad et al., 2023*). Raw Micro-C sequencing reads were downloaded from GEO (GSE206131) and analyzed with the pipeline we used for MChIP-C and PLAC-seq data processing. We once again relied on MChIP-C-defined viewpoints to extract merged promoter-anchored Micro-C profiles and viewpoint-specific Micro-C profiles. The data was binned in 100 bp or 250 bp bins to plot viewpoint-specific interaction profiles. In all other types of analysis we used fixed 250 bp bins.

We also created all-to-all Micro-C proximity ligation matrices using conventional distiller-nf pipeline (v. 0.3.4; https://github.com/open2c/distiller-nf). To generate iteratively balanced promoter-anchored Micro-C profiles, we dumped 250 bp binned cooler matrix to a text stream ('cooler dump' command) and then selected the balanced signal from the bins residing within MChIP-C viewpoints. We then aggregated this signal into a merged genome-wide bedgraph file and converted it to a bigwig profile using bedgraphToBigwig UCSC utility.

To identify point interactions in the Micro-C dataset, we used Mustache algorithm (v 1.0.1; *Roayaei Ardakany et al., 2020*). We called interactions in balanced contact matrices at resolutions of 250 bp, 500 bp, 1 kb, 2 kb, 5 kb, and 10 kb using -pt 0.1 and -st 0.88 options. We combined interactions called at all resolutions in one set; if an interaction was detected at multiple resolutions, we retained a variant with coarser boundaries and discarded finer ones.

## Hi-C data reanalysis

Raw Hi-C sequencing reads were downloaded from GEO (GSE63525) and all-to-all Hi-C proximity ligation matrices were generated with the distiller-nf pipeline. To obtain raw and iteratively balanced promoter-anchored Hi-C profiles, we dumped 1 kb binned cooler matrix to a text stream ('cooler dump' command) and then selected both raw and balanced signals from the bins residing within MChIP-C viewpoints. We then aggregated these signals into two separate merged genome-wide bedgraph files (one containing raw signal and another containing balanced signal) and converted them to bigwig profiles with bedGraphToBigwig UCSC utility.

To identify point interactions in the Hi-C dataset, we used Mustache algorithm. We called interactions in balanced contact matrices at resolutions of 1 kb, 2 kb, 5 kb, 10 kb, and 20 kb using -pt 0.1

and -st 0.88 options. We combined interactions called at all resolutions into one set; if an interaction was detected at multiple resolutions, we retained a variant with coarser boundaries and discarded finer ones.

## Comparison of merged MChIP-C, PLAC-seq, promoter-anchored Micro-C and promoter-anchored Hi-C signals around distal DNase hypersensitive sites

DNase sensitivity, CTCF ChIP, H3K4me3 ChIP, merged MChIP-C, merged PLAC-seq and merged promoter-anchored Micro-C and Hi-C signals spanning 5 kb regions flanking distal CTCF sites and CTCF-less DHSs were plotted as heatmaps and average profiles with deeptools (*Figure 2b*). Distal CTCF sites with the peak score (the 5th column of CTCF peak file) less than 250 and DHS sites with the signalValue (the 7th column of DHS peak file) less than 200 were excluded from the analysis. In all subsequent analysis concerning DHSs, we used the same signalValue cutoff.

To plot violin graphs for proximity ligation signal in and around distal DHSs, we used 250 bp binned signal matrices generated by deeptools as intermediaries during heatmap generation. For each proximity ligation method, we compiled a set of signal bins and two sets of background bins, 1 kb and 5 kb away from the peak centers. For the signal set, we carved out from the matrix a pair of central columns corresponding to bins surrounding each individual analyzed distal DHS. For each background set, we selected 2 columns corresponding to bins separated by either 1 or 5 kb from the center of DHSs (1 column from each side of the DHSs). We defined the average enrichment of the signal in CTCF sites and CTCF-less DHS for each method as the ratio between median signal in the center and median signal in the 5 kb background bin. To assess the resolution of the methods, we also calculated the ratio between the median signal in the 1 kb background bin and the median signal in the 5 kb background bin.

## Comparison of promoter-DHS and promoter-CTCF interactions identified in MChIP-C, PLAC-seq and Micro-C data

For an even more detailed comparison of MChIP-C data to the data generated with the previously described protocols (Micro-C and PLAC-seq), we focused on the promoter-centered interactions identified with each method in K562 cells. First, we selected unique promoter-CTCF (P-CTCF) and promoter-DHS (P-DHS) interactions by overlapping promoter-interacting regions detected with each of these three techniques individually, with the sets of distal CTCF sites and distal CTCF-less DHS sites. For each interacting pair in each method, we calculated the distance between the center of the DHS or CTCF site and the center of the overlapping promoter interacting region and analyzed the distribution of such distances for each method (*Figure 2—figure supplement 2a*, violin plots). Then, we compiled a non-redundant list of P-DHS interactions identified by at least one of the methods, and split this combined list into seven groups in accordance with the ability of each individual P-DHS pair to be identified by a certain set of methods (*Figure 2—figure supplement 2a*, Venn diagrams). We also performed the same analyses for P-CTCF interactions.

Next, we focused on 3 specific groups of interactions: common (identified in all 3 datasets), PLAC-seq-specific, and Micro-C-specific (*Figure 2—figure supplement 2b-d*). To visualize MChIP-C, PLAC-seq and Micro-C signals around promoter-interacting DHS and CTCF-sites in these 3 groups, we plotted 250 bp binned signal of ligation to the corresponding anchor promoters as heatmaps and average profiles. For these same sets of P-DHS and P-CTCF pairs, we also plotted the distribution of proximity ligation signals in the centers of the sites and in two different sets of background bins (1 kb and 5 kb apart from the centers of the sites). We used normalized MChIP-C signal, raw PLAC-seq and Micro-C signals in this analysis. For average profiles and violin plots, we also included ICE-normalized Micro-C signal. Next, we defined the average enrichment of the signal in each group of sites for each method as the ratio between mean signal in the center and mean signal in the 5 kb background bin. To assess the resolution of the methods, we also calculated the ratio between the mean signal in the 1 kb background bin and the mean signal in the 5 kb background bin.

## Analysis of CTCF motif orientation in viewpoints and PIRs

The presence of CTCF motifs in viewpoints and PIR bins, as well as their orientation, were determined with HOMER (v4.11.1; *Heinz et al., 2010*) ('annotatePeaks.pl'). To compare MChIP-C-identified

CTCF-based interactions with Hi-C loops, we used the list of 6,057 K562 Hi-C loops identified by *Rao et al., 2014*.

To evaluate the number and orientation of CTCF-motifs in randomized sets of PIRs, we performed 100 independent random permutations, changing the positions of the PIRs while preserving P-PIR distance distribution. Then we assessed the presence and orientation of CTCF motifs in each generated set of PIRs and averaged the obtained results to get the number and distribution of motifs per randomization.

## Transcription related factor enrichment analysis in PIRs

To assess transcription-related factor (TRF) enrichment in PIRs, we exploited a compendium of publicly available K562 ChIP-seq profiles (see *Supplementary file 1* for a complete list of ChIP-seq datasets used in the present study). The majority of the analyzed datasets were downloaded from the ENCODE portal (268 out of 271). CDK8 (*Pelish et al., 2015*; CistromeDB: 56379) and MED1 (CistromeDB: 74667) occupancy data were downloaded from the Cistrome database, BRD4 (*Liu et al., 2017*) – from GEO (GSM2635249). Hg38 datasets were lifted to hg19 with CrossMap (v.0.6.0; *Zhao et al., 2014*). All ChIP-seq TRF peaks with the signalValue (the 7th column of each peak file) less than 25 were excluded from all the TRF-related analysis.

First, we overlapped MChIP-C PIRs with TRF binding sites. To estimate relative enrichment, we calculated the expected number of overlaps. We did this for each individual TRF by averaging the overlap counts in 100 independent random permutations, changing the positions of the PIRs while preserving P-PIR distance distribution. The position of PIRs in each permutation was established by random reshuffling of distances separating viewpoints from their original PIRs. Enrichment of TRF binding was then calculated as log2 of the ratio between the experimentally observed and the expected number of overlaps (*Supplementary file 1*; *Figure 3b*). Enrichment of TRF motifs in PIRs was evaluated with the HOMER package ('findMotifsGenome.pl' command).

## PIR-overlapping DHS clustering and cluster annotation

To separate 19,129 PIR-overlapping DHSs (promoter interacting DHSs) into biologically-relevant groups according to their TRF binding, we used hierarchical clustering with binary features reflecting the binding status of the analyzed TRFs. Here we used only 164 TRFs bound to more than 1,000 promoter interacting DHSs and PIR enrichment higher than 2 (*Figure 3—figure supplement 2a*). We used the Ward's Minimum Variance distance metric for clustering and chose a cutoff of four clusters.

For each cluster, we calculated the number of DHSs bound by each of the analyzed TRFs. Next, for each cluster we identified 20 TRFs which were bound to the greatest number of DHSs. Then we merged these four sets of TRFs into one list which was comprised of 27 unique TRFs. We plotted the binding of these selected TRFs in PIR-overlapping DHSs as a heatmap (*Figure 3c*). To characterize the identified clusters, we overlapped cluster-assigned DHSs with K562 ChromHMM states (*Ernst et al., 2011*). We also overlapped DHSs with CRISPRi-verified enhancers from *Fulco et al., 2019* and *Gasperini et al., 2019* in order to assess enhancer enrichment of each cluster with a one-sided binomial test.

To calculate the number and distance distributions of MChIP-C interactions associated with the identified DHS clusters (*Figure 2—figure supplement 2b*), we overlapped the complete set of P-PIR interactions with positions of the promoter-interacting DHSs.

## Random forest regression for modeling of MChIP-C signal

We used a random forest regression model (R library 'ranger'; *Wright and Ziegler, 2017*) to predict normalized MChIP-C signal and to identify predictive features. We focused on a subset of viewpoint-DHS pairs separated by <250 kb. We identified 107,570 such viewpoint-DHS pairs with a non-zero count of MChIP-C reads and filtered out two outliers (chr16:214,895–215,045 and chr5:693,455–693,605) which were 175 and 82 IQRs above the third quartile. For training the initial model, we used the following features: viewpoint-DHS distance, CTCF motif orientation on both the viewpoint and DHS and CTCF ChIP-seq signal on both the viewpoint and DHS. Training parameters were default with mtry = 3, num.trees=100.

Next, we used a greedy forward feature selection approach to add 10 features out of a set of 270 TRF ChIP-seq signals on the DHSs (*Figure 3d*; *Figure 3—figure supplement 2c*). At each feature

selection step, we evaluated all features and added the most predictive feature to the model. To assess the predictive power of a feature, we retrained the model with the feature and used threefold cross-validation to evaluate either $R^2$ or AUC (where MChIP-C was partitioned as higher/lower than the median). To assess reproducibility of the greedy feature selection, we repeated the entire procedure three times.

## Estimating correlation between chromatin binding of EP300 and SWI/SNF-complex

To compare chromatin binding profiles of histone acetyltransferase EP300 and of the ARID1B, DPF2, SMARCE1 subunits of SWI/SNF-complex in K562 cells, we first calculated ChIP-seq read coverage in K562 DHSs for each of the proteins with deeptools ('multiBigwigSummary BED-file' command). Then we calculated Pearson's correlation between EP300 coverage and coverages of each of the studied SWI/SNF subunits.

## Downsampling of MChIP-C data and analysis of the downsampled datasets

For a more thorough comparison of MChIP-C with other C-methods in terms of their ability to detect increased ligation frequency between distal regulatory elements and their target promoters, we downsampled generated MChIP-C dataset. We wanted to decrease the total number of unique distal pairs mapping to the viewpoints in the downsampled MChIP-C dataset so that it would be equal to the number of such reads in the complete PLAC-seq and Micro-C datasets. First, we evaluated the numbers of such pairs for all three approaches. We found that PLAC-seq and Micro-C datasets contained approximately two times less useful pairs than complete MChIP-C dataset (~19.75 million and ~23.15 million vs ~42.7 million). Thus, we sampled 50% of raw reads from each of the 4 biological replicates of MChIP-C ('pairtools sample' command). The downsampled dataset contained ~24.86 million useful pairs. We performed interaction calling in the downsampled MChIP-C dataset with the exact same pipeline we used previously for the MChIP-C datasets from individual replicates. Then, we downsampled MChIP-C data even further to 25% (~13.95 million useful pairs) and 10% (~6.04 million useful pairs) of the original read depth and again performed the interaction calling procedure used previously. Note that the number of raw reads in the 10% downsampled dataset (~354 million) was close to the number of raw reads in the PLAC-seq experiment (~314 million) and approximately ten times lower than in the Micro-C dataset we used for the comparison (~3,229 million).

To assess the robustness of the MChIP-C interactions to downsampling, we first classified identified loops into P-DHS and P-CTCF classes as described previously (see the definition of P-DHS and P-CTCF interactions above). Then, we pooled all P-DHS interactions identified in the full dataset and in all downsampled datasets into a combined non-redundant set. We also performed similar pooling with P-CTCF loops. To visualize the reproducibility of the interaction calls, we plotted the resulting data as UpSet plots using UpSetR (v.1.4.0; *Conway et al., 2017*) R package (*Figure 4—figure supplement 1d*).

## Comparing MChIP-C interactions with functionally verified E-P pairs

We used K562 CRISPRi datasets from *Fulco et al., 2019* and *Gasperini et al., 2019* to compile a list of 366 verified E-P pairs and 54,811 non-regulatory DHS-P pairs (*Supplementary file 1*). In order to directly compare these pairs to MChIP-C identified chromatin interactions, we included in this list only pairs with promoters overlapping MChIP-C viewpoints and pairs with distal sites within 5 kb-1Mb of the promoter.

Then we overlapped the obtained E-P and DHS-P pairs with interactions identified with various C-methods: MChIP-C (full and downsampled versions), PLAC-seq, Micro-C and Hi-C. Before overlapping we extended PIRs of the identified MChIP-C interactions by 500 bp in both directions, and loop anchors of the identified Micro-C and Hi-C interactions by 2500 bp in both directions. Using the obtained results, we separated E/DHS-P pairs in four groups for each analyzed method: DHS-P pairs lacking interactions, DHS-P pairs with underlying interactions, E-P pairs lacking interactions and E-P pairs with underlying interactions. We excluded all CTCF-bound E/DHS sites and visualized normalized MChIP-C signal as well as raw PLAC-seq, Micro-C and Hi-C signal around each individual E/DHS site (reflecting proximity between the site and the paired promoter) as a heatmap (*Figure 4a* and

*Figure 4—figure supplement 1a*). Finally, we plotted average MChIP-C (normalized), PLAC-seq (raw), Micro-C (raw and balanced) and Hi-C signal (raw and balanced) for each of the four E/DHS-P groups as averaged profiles. (*Figure 4a* and *Figure 4—figure supplement 1a*). Micro-C heatmap pile-ups around E/DHS-P pairs (*Figure 4—figure supplement 1a*) were aggregated based on the genome-wide matrices with cooltools (v.0.5.1; *Abdennur et al., 2024*).

We calculated sensitivity (recall), precision and false positive rate for the predictors based on the presence of interactions between promoters and their potential regulatory elements (E/DHS) identified with each of the analyzed C-methods (*Figure 4—figure supplement 1b*). As a reference, we used predictors based on the assignment of each DHS to the closest active gene (TPM >0.5, from ENCODE ENCFF934YBO dataset) as its enhancer or the assignment of each DHS to the two closest active genes as their enhancer. Also as a reference we built a precision-recall and receiver operating characteristic curves for a predictor based on the thresholds inversely proportional to the genomic distance between the DHS/enhancer and TSS of the potential target gene.

## Data processing and visualization

The following software and packages were used for analysis and visualization: bwa (v.0.7.17), Bowtie2 (v.2.3.4), pairtools (v.0.3.0), samtools (v.1.15.1), bedtools (v2.26.0), deeptools (v.3.5.1), MACS2 (v.2.2.7.1), Python (v.3.7.12; numpy v.1.21.6, pandas v.1.3.5, matplotlib v.3.5.3, cooler v.0.9.1, cooltools v.0.5.1), R (v. 4.2.1; dplyr v.1.0.9, tidyr v.1.2.0, ggplot2 v.3.3.6, gplots v. 3.1.3, data.table v.1.14.8, GenomicRanges v.1.48.0, reshape2 v.1.4.4, fitdistrplus v.1.1–8, RColorBrewer v.1.1–3, dendextend v.1.17.11.16.0, dendroextras v.0.2.3, GGally v.2.1.2, gridExtra v.2.3, eulerr v.7.0.1, ranger v.0.16.00.14.1, caret v.6.0–93, PRROC v.1.3.1, UpSetR v.1.4.0), HOMER (v.4.11.1), CrossMap (v.0.6.0), nextflow (v.22.10.4.5836), ditiller-nf pipeline (v.0.3.4), Mustache (v.1.3.2), UCSC utilities (bedGraphToBigWig, liftOver, wigToBigWig), Integrative Genomics Viewer (v.2.8.0), Adobe Illustrator (v.23.0.1).

## Acknowledgements

This work was supported by the Russian Science Foundation (21-64-00001), as well as Technion funding for AKG. MChIP-C experiments were performed using the equipment of IGB RAS facilities, supported by the Russian Ministry of Science and Higher Education. We thank Hagai Kariti and David Cohen for computational support.

## Additional information

### Funding

| Funder | Grant reference number | Author |
| --- | --- | --- |
| Technion - Israel Institute of Technology, RBNI - Russell Berrie Nanotechnology Institute | | Arkadiy K Golov Noam Kaplan |
| Russian Science Foundation | 21-64-00001 | Alexey A Gavrilov Sergey V Razin |

The funders had no role in study design, data collection and interpretation, or the decision to submit the work for publication.

### Author contributions

Arkadiy K Golov, Conceptualization, Software, Formal analysis, Investigation, Visualization, Methodology, Writing – original draft, Writing – review and editing, Experiments; Alexey A Gavrilov, Sergey V Razin, Conceptualization, Supervision, Funding acquisition, Investigation, Methodology, Project administration, Writing – review and editing; Noam Kaplan, Formal analysis, Supervision, Funding acquisition, Investigation, Methodology, Writing – original draft, Project administration, Writing – review and editing

Author ORCIDs
Arkadiy K Golov ⓘ https://orcid.org/0000-0001-7802-3163
Noam Kaplan ⓘ https://orcid.org/0000-0001-9940-1987
Sergey V Razin ⓘ https://orcid.org/0000-0003-1976-8661

Reviewer #1 (Public review): https://doi.org/10.7554/eLife.91596.3.sa1
Reviewer #2 (Public review): https://doi.org/10.7554/eLife.91596.3.sa2
Reviewer #3 (Public review): https://doi.org/10.7554/eLife.91596.3.sa3
Author response https://doi.org/10.7554/eLife.91596.3.sa4

## Additional files

### Supplementary files
• Supplementary file 1. MChIP-C processed datasets and statistics.
• MDAR checklist

### Data availability
MChIP-C and ChIP-seq sequencing data as well as derivative genomic profiles and contact matrices have been deposited in GEO under accession code GSE225087. Code to reproduce analysis is available on GitHub https://github.com/KaplanLab/MChIP-C/ (copy archived at *KaplanLab, 2023*). Interactive MChIP-C interaction profiles of individual promoters are browseable at https://storage.googleapis.com/arkadiy/static/index.html .

The following dataset was generated:

| Author(s) | Year | Dataset title | Dataset URL | Database and Identifier |
|---|---|---|---|---|
| Golov AK, Gavrilov AA, Kaplan N, Razin SV | 2023 | Genome-wide nucleosome-resolution map of promoter-centered interactions in human cells corroborates the enhancer-promoter looping model | https://www.ncbi.nlm.nih.gov/geo/query/acc.cgi?acc=GSE225087 | NCBI Gene Expression Omnibus, GSE225087 |

The following previously published datasets were used:

| Author(s) | Year | Dataset title | Dataset URL | Database and Identifier |
|---|---|---|---|---|
| Chen BP, Ren B | 2021 | Systematic Discovery and Therapeutic Targeting of Pro-growth Enhancers in Human Cancer Cells [PLAC-seq] | https://www.ncbi.nlm.nih.gov/geo/query/acc.cgi?acc=GSE161873 | NCBI Gene Expression Omnibus, GSE161873 |
| Barshad G, Danko CG | 2022 | RNA polymerase II and PARP1 shape enhancer-promoter contacts [Micro-C] | https://www.ncbi.nlm.nih.gov/geo/query/acc.cgi?acc=GSE206131 | NCBI Gene Expression Omnibus, GSE206131 |
| Rao S, Huntley M, Lieberman Aiden E | 2014 | A three-dimensional map of the human genome at kilobase resolution reveals prinicples of chromatin looping | https://www.ncbi.nlm.nih.gov/geo/query/acc.cgi?acc=GSE63525 | NCBI Gene Expression Omnibus, GSE63525 |

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
