## [Editor Report · eLife Assessment]

Identifying chromatin interactions with high sensitivity and resolution at the genome-wide scale continues to be technically challenging. This study introduces findings based on the improved MNase-based proximity ligation method, MChIP-C, which enables genome-wide measurement of chromatin interactions at single-nucleosome resolution. The evidence presented in this manuscript is **convincing**, and the technological advancements will be **valuable** for the study of 3D genome architecture.

---

## [Referee Report · Reviewer #1 (Public review)]

The authors presented a new MNase-based proximity ligation method called MChIP-C, allowing for the measurement of protein-mediated chromatin interactions at single-nucleosome resolution on a genome-wide scale. With improved resolution and sensitivity, they explored the spatial connectivity of active promoters and identified the potential candidates for establishing/maintaining E-P interactions. Finally, with published CRISPRi screens, they found that most functionally-verified enhancers do physically interact with their cognate promoters, supporting the enhancer-promoter looping model.

While the study's experimental approach and findings are interesting. However, several issues need to be addressed:

(1) The authors described that "the lack of interaction between experimentally-validated enhancers and their cognate promoters in some studies employing C-methods has raised doubts regarding the classical promoter-enhancer looping model", so it's intriguing to see whether the MChIP-C could indeed detect the E-P interactions which were not identified by C-methods as they mentioned (Benabdallah et al., 2019; Gupta et al., 2017). I agree that they identified more E-P interactions using MChIP-C, but specifically, they should show at least 2-3 cases. It's important since this is the main conclusion the authors want to draw.

(2) The authors compared their data to those of Chen et al. (Chen et al., 2022), who used PLAC-seq with anti-H3K4me3 antibodies in K562 cells and standard Micro-C data previously reported for K562, concluding that "MChIP-C achieves superior sensitivity and resolution compared to C-methods based on standard restriction enzymes.". This is not convincing since they only compared their data to one dataset. More datasets from other cell lines should be included.

(3) The reasons to choose Chen's data (Chen et al., 2022) and CRISPRi screens (Fulco et al., 2019; Gasperini et al., 2019) should be provided since there are so many out there.

(4) The authors identify EP300 histone acetyltransferase and the SWI/SNF remodeling complex as potential candidates for establishing and/or maintaining enhancer-promoter interactions, but not RNA polymerase II, mediator complex, YY1 and BRD4. More explanation is needed for this point since they're previously suggested to be associated with E-P interactions.

(5) The limitations of the method should be discussed.

---

## [Referee Report · Reviewer #2 (Public review)]

Summary:

Golov et al has performed the capture MChIP-C using H3K4me3 antibody. The new method significantly increases the resolution of Micro-C and can detect the clear interactions which is not well described in the previous HiChIP/PLAC-seq method. Overall, the paper represented a significant technological advance which can be valuable to the 3D genomic field in the future.

The authors have addressed all my concerns and comments.

---

## [Referee Report · Reviewer #3 (Public review)]

Summary:

This manuscript represents a technology development- specifically an micrococcal nuclease chromatin capture approach, termed MChIP-C to identify promoter centered chromatin interactions at single nucleosome resolution via a specific protein, similar to HiChIP, ChIA-PET, etc.. In general the manuscript is technically well done.

Strengths:

Methods appear to hold promise to improve both the sensitivity and resolution of protein-centered chromatin capture approaches.

Weaknesses:

Downsampling analysis gives a better idea of the strengths of the approach, especially related to individual loci. While this method does outperform other approaches, it remains technically sophisticated and for some labs may not be worth the additional effort for the increase in information. Also, until tested and proven by other groups, it is difficult to know how impactful this approach will be.

---

## [Author Response]

The following is the authors’ response to the original reviews.

We thank the reviewers for their overall positive evaluation of the manuscript and finding MChIP-C to be a valuable technological advance. To address the reviewer’s helpful comments and recommendations, we performed several additional analyses and improved the text and figures.

Briefly, we extended and clarified the main text and methods, added analyses of interactions at consensus and method-specific CTCF/DHS sites (Figure S3), added additional comparison tracks to other methods in specific loci (Figure 4), added examples of MChIP-C E-P interactions at previously-verified loci (Figure S2a) and added extensive MChIP-C downsampling analysis (Figure S6).

**Recommendations for authors:**

**Reviewer #2 Recommendations For The Authors:**
(1) Provide .HiC and .cool files for the community to explore the data.

We thank the reviewer for this suggestion. We have uploaded both the raw and processed data to GEO. We note that .cool and .hic formats may be less useful for this type of data, since it includes only promoter-based interactions and thus the resulting interaction matrix is extremely sparse at the relevant resolutions. In addition, we provide an online genomic browser for our data.

(2) Provide an R or bioconda package for future data processing.

We thank the reviewer for this suggestion. We have organized and streamlined the relevant code for processing MChIP-C data and it is available as a github repository.

(3) The authors should avoid using "mln" for "million".

We thank the reviewer for this suggestion. We have corrected this in the text.

**Reviewer #3 (Recommendations For The Authors):**
(1) Figure 2- A handful of sites identified by MChIP-C should be verified by 3C or 4C to validate they are true interactions using an orthogonal approach.

We thank the reviewer for this suggestion. As we show in the current manuscript (and supported by several papers using MNase-based C-methods), C-methods based on restriction enzymes are considerably less sensitive than those based on MNase, so using these methods for anecdotal validation may not be adequate. In addition, it is difficult to extract accurate quantitative measurements from 3C and 4C due to challenges in bias normalization. As a large-scale alternative, we analyzed a set of consensus promoter-CTCF and promoter-DHS interactions identified by all 3 methods (PLAC-seq/Micro-C/MChIP-C; Figure S3). We find that MChIP-C shows clearly superior resolution and sensitivity on these consensus sites. In fact, even for sites which were only called by one of the competing methods, we still see better signal in the MChIP-C data (suggesting that our simplistic MChIP-C peak-calling approach could be improved for further gain). However, as this analysis focuses on “easily detectable” consensus sites, we also emphasize the importance of inspecting interactions which are not detected clearly by alternative methods. To this end, we now show in our manuscript interaction profiles for 11 loci (MYC, PTGER3, CITED2, BTG1, ANTXR2, SEMA7A, LMO2, GATA1, HBG2, VEGFA, MYB), each showing high-resolution MChIP-C interactions which coincide with expected genomic features (p300, CTCF, H3K27ac, known enhancers) and are not clearly observable in Micro-C and PLAC-seq. We also note that the extended overlap of detected MChIP-C interactions with functionally validated enhancers (as measured by CRISPRi) provides an additional large-scale orthogonal validation.

(2) A supplemental table indicating read pair depth, etc, similar to S02, should be added for the datasets used for comparison (HiChIP-etc). Given the age differences between some of the reference data used, it may represent simply an improvement by increasing sequencing depth rather than a true technical advantage.

We thank the reviewer for this suggestion. We have added the sequencing depths of the relevant datasets in the methods section. We also performed extensive downsampling analyses as explained in response to the next point.

(3) I would recommend performing a downsampling analysis to determine at what point the MChIP-C data reaches saturation in terms of the number of reads, with a comparison to the HiChIP reference data. This would allow a more objective measure of the sensitivity of the assays with reference to read depth.

We thank the reviewer for this suggestion. First, we note that downsampling does not affect the high sensitivity and resolution results as shown in aggregate plots (e.g. Figure 2 and Figure S3). However, downsampling can affect individual peak calling. We thus downsampled our data to 50%, approximately matching the number of total informative reads of both PLAC-seq and Micro-C (i.e. ~20M). We also further downsampled our data to 25% and 10%. With respect to prediction of K562 functionally validated enhancer-promoter interactions (Figure S6b), even at 25% downsampling MChIP-C achieves both a higher recall and higher precision than the other methods, with a slightly higher false-positive rate. At 10% sampling, recall is slightly worse than Micro-C and PLAC-seq, but both the precision and false-positive rate are better than the alternatives. With respect to saturation, we plotted the number of unique distal cis read pairs versus the total number of reads (Figure S6c), and find that our MChIP-C data does not yet show saturation. We also show that downsampling our data to 50% maintains ~80% of the called interactions (Figure S6d).

(4) "our results suggest that MChIP-C achieves superior sensitivity and resolution compared to C-methods based on standard restriction enzymes." The sensitivity claims are supported by Figure 2, but not the resolution claims. This is particularly challenging when using histone marks since they can be broad. To directly compare the resolution of MChIP-C to other approaches such as ChIA-PET or HiChIP CTCF or a similar DNA binding protein is required.

We thank the reviewer for this suggestion. We first note that actually both sensitivity and resolution are relevant for the results shown in Figure 2 and for the signal-to-noise calculations. This is because the low resolution of PLAC-seq peaks can result in very broad peaks that cover the entire area of the interrogated window (5kb on each side), which could seem like low sensitivity. However, we believe that the new Figure S3 may show the higher resolution of MChIP-C more clearly, as do the 11 locus interaction profiles tracks shown in Figure 2, Figure 4 and Figure S2.

**Public reviews:**

**Reviewer #1:**
The authors presented a new MNase-based proximity ligation method called MChIP-C, allowing for the measurement of protein-mediated chromatin interactions at single-nucleosome resolution on a genome-wide scale. With improved resolution and sensitivity, they explored the spatial connectivity of active promoters and identified the potential candidates for establishing/maintaining E-P interactions. Finally, with published CRISPRi screens, they found that most functionally verified enhancers do physically interact with their cognate promoters, supporting the enhancer-promoter looping model.The study's experimental approach and findings are interesting. However, several issues need to be addressed.(1) The authors described that "the lack of interaction between experimentally-validated enhancers and their cognate promoters in some studies employing C-methods has raised doubts regarding the classical promoter-enhancer looping model", so it's intriguing to see whether the MChIP-C could indeed detect the E-P interactions which were not identified by C-methods as they mentioned (Benabdallah et al., 2019; Gupta et al., 2017). I agree that they identified more E-P interactions using MChIP-C, but specifically, they should show at least 2-3 cases. It's important since this is the main conclusion the authors want to draw.

We thank the reviewer for this suggestion. As we show in the current manuscript (and supported by several papers using MNase-based C-methods), C-methods based on restriction enzymes are considerably less sensitive than those based on MNase, so using these methods for anecdotal validation may not be useful. In addition, it is difficult to extract accurate quantitative measurements from 3C and 4C due to challenges in bias normalization. As a large-scale alternative, we analyzed a set of consensus promoter-CTCF and promoter-DHS interactions identified by all 3 methods (PLAC-seq/Micro-C/MChIP-C; new Figure S3). We find that MChIP-C shows clearly superior resolution and sensitivity on these consensus sites. However, as this analysis focuses on “easily detectable” consensus sites, we also emphasize the importance of inspecting interactions which are not detected clearly by alternative methods. To this end, we now show in our manuscript interaction profiles for 11 loci (MYC, PTGER3, CITED2, BTG1, ANTXR2, SEMA7A, LMO2, GATA1, HBG2, VEGFA, MYB), each showing high-resolution MChIP-C interactions which coincide with expected genomic features (p300, CTCF, H3K27ac, known enhancers) and are not clearly observable in Micro-C and PLAC-seq. We also note that the extended overlap of detected MChIP-C interactions with functionally validated enhancers (as measured by CRISPRi) provides an additional large-scale orthogonal validation.

(2) The authors compared their data to those of Chen et al. (Chen et al., 2022), who used PLAC-seq with anti-H3K4me3 antibodies in K562 cells and standard Micro-C data previously reported for K562, concluding that "MChIP-C achieves superior sensitivity and resolution compared to C-methods based on standard restriction enzymes.". This is not convincing since they only compared their data to one dataset. More datasets from other cell lines should be included.

We thank the reviewer for this suggestion. We would like to clarify that all datasets in the paper are K562 datasets, and this cell line is unique in the availability of CRISPRi screens, PLAC-Seq, Micro-C, and hundreds of ChIP-Seq tracks for it. We would expect datasets from other cell types to have changes in their regulatory interactions, so they would be less adequate for direct comparison. In addition, the general resolution and sensitivity limitations (e.g. due to restriction fragment size) are not dependent on cell type and has been shown in other MNase-based method papers.

(3) The reasons for choosing Chen's data (Chen et al., 2022) and CRISPRi screens (Fulco et al., 2019; Gasperini et al., 2019) should be provided since there are so many out there.

We thank the reviewer for this comment. We selected these CRISPRi screen datasets since they match the cell type (K562) which we used for MChIP-C, and we selected the PLAC-seq data as it is the only PLAC-seq/HiChIP dataset which matches both the cell type (K562) and the antibody (H3K4me3).

(4) The authors identify EP300 histone acetyltransferase and the SWI/SNF remodeling complex as potential candidates for establishing and/or maintaining enhancer-promoter interactions, but not RNA polymerase II, mediator complex, YY1, and BRD4. More explanation is needed for this point since they're previously suggested to be associated with E-P interactions.

We thank the reviewer for this comment. We apologize for this point being unclear: as Figure S5 shows, we actually did identify Pol2, mediator YY1 and BRD4 as predictive features, but P300 and SWI/SNF show somewhat higher predictive power. We have now clarified this in the text.

(5) The limitations of the method should be discussed.

We thank the reviewer for this suggestion. We have now added to the text a discussion of what we view as the current main limitation of the method, namely its low fraction of informative reads.

**Reviewer #2:**
Summary:Golov et al performed the capture of MChIP-C using the H3K4me3 antibody. The new method significantly increases the resolution of Micro-C and can detect clear interactions which are not well described in the previous HiChIP/PLAC-seq method. Overall, the paper represents a significant technological advance that can be valuable to the 3D genomic field in the future.Strengths:(1) The authors established a novel method to profile the promoter center genomic interactions based on the Micro-C method. Such a method could be very useful to dissect the enhancer promoter interaction which has long been an issue for the popular HiC method.(2) With the MChIP-C method the authors are able to find new genomic interactions with promoter regions enriched in CTCF. The author has significantly increased the detection sensitivity of such methods as PLAC-seq, Micro-C, and HiChIP.(3) The authors identified a new type of interaction between the CTCF-less promoter and the CTCF binding site. This particular type of interaction could explain the CTCF's function in regulating gene transcription activity as observed in many studies. I personally think the second stripe model of P-CTCF interaction is more likely as this has been proposed for the super-enhancer stripe model before. The author should also discuss this part of the story more.Weaknesses:(1) The data presentation should include the contact heat map. The current data presentation makes it hard for the readers to have a comprehensive view of pair-wise interactions between promoters and the PIR. In particular, these maps may directly give answers to the proposed model of promoter-CTCF interactions by the authors in Figure 3a.

We thank the reviewer for this suggestion. We note that since the data mainly includes promoter-based interactions, the resulting interaction matrix is extremely sparse at the relevant resolutions. Specifically with respect to promoter-CTCF interactions, without a good sampling of the entire interaction matrix it is difficult to confidently distinguish between the two models only based on MChIP-C data, as it would require data about interaction between non-promoter regions and CTCF.

(2) In Fig 3D, there seems a very limited increase of power predicting MChIP-C signal for DHS-promoter pairs beyond the addition of CTCF. This figure could be simplified with fewer factors.

We thank the reviewer for this suggestion. We agree that the last factors do not add predictive power, but we do not think this overly complicates the figure and we prefer to leave these for the reader to evaluate.

(3) The current method seems to have a big fraction of unusable reads. How the authors process the data should be included to allow for future reproduction. Ideally, the authors should generate a package on R or Bioconda for this processing.

We thank the reviewer for this suggestion. We agree that the fraction of informative reads is small with respect to some other methods, and expect future versions of MChIP-C to address this limitation. We have organized and streamlined the relevant code for processing MChIP-C data and it is available as a github repository.

**Reviewer #3:**
Summary:This manuscript represents a technological development- specifically a micrococcal nuclease chromatin capture approach, termed MChIP-C to identify promoter-centered chromatin interactions at single nucleosome resolution via a specific protein, similar to HiChIP, ChIA-PET, etc.. In general, the manuscript is technically well done. Two major issues raise concerns that need to be addressed. First, it does not appear that novel chromatin interactions identified by MChIP-C which were missed by other approaches such as HiChIP, were validated. This is central to the argument of "improved" sensitivity, which is one of the key factors to assess sensitivity. Second is the question of resolution. Because the authors focus on a histone mark (H3K4me3) it is unclear whether the resolution of the assay truly exceeds other approaches, especially microC. These two issues are not completely supported by the data provided.Strengths:The method appears to hold promise to improve both the sensitivity and resolution of protein-centered chromatin capture approaches.Weaknesses:(1) Specific validation experiments to demonstrate the identification of previously missed novel interactions are missing.

We thank the reviewer for this suggestion. Given that such interactions are missed by Micro-C and PLAC-seq, it would not make sense to use these methods for validation. We thus propose that MChIP-C interactions can be validated by their overlap with expected genomic features. To this end, we now show in our manuscript interaction profiles for 11 loci (MYC, PTGER3, CITED2, BTG1, ANTXR2, SEMA7A, LMO2, GATA1, HBG2, VEGFA, MYB), each showing high-resolution MChIP-C interactions which coincide with expected genomic features (p300, CTCF, H3K27ac, known enhancers) and are not clearly observable in Micro-C and PLAC-seq. In addition, the higher overlap of MChIP-C interactions with functionally-validated K562 enhancer-promoter interactions (provided by CRISPRi screens) provides further functional validation for novel MChIP-C interactions.

(2) It is unclear if the resolution is really superior based on the data provided.

We thank the reviewer for this comment. We first note that actually both sensitivity and resolution are relevant for the results shown in Figure 2 and for the signal-to-noise calculations. This is because the low resolution of PLAC-seq peaks can result in very broad peaks that cover the entire area of the interrogated window (5kb on each side), which could seem like low sensitivity. However, we believe that the new Figure S3 may show the higher resolution of MChIP-C more clearly, as do the 11 locus interaction profiles tracks shown in Figure 2, Figure 4 and Figure S2.

(3) It is unclear how much advantage the approach has, especially compared to existing approaches such as HiChIP since sequencing depth as a variable is not adequately addressed.

We thank the reviewer for this comment. First, we note that downsampling does not affect the high sensitivity and resolution results as shown in aggregate plots (e.g. Figure 2 and Figure S3). However, downsampling can affect individual peak calling. We thus downsampled our data to 50%, approximately matching the number of total informative reads of both PLAC-seq and Micro-C (i.e. ~20M). We also further downsampled our data to 25% and 10%. With respect to prediction of K562 functionally validated enhancer-promoter interactions (Figure S6b), even at 25% downsampling MChIP-C achieves both a higher recall and higher precision than the other methods, with a slightly higher false-positive rate. At 10% sampling, recall is slightly worse than Micro-C but both the precision and false-positive rate are better than the alternatives.